# A self-balancing circuit centered on MoOsm1 kinase governs adaptive responses to host-derived ROS in *Magnaporthe oryzae*

Xinyu Liu[1], Qikun Zhou[1], Ziqian Guo[1], Peng Liu[1], Lingbo Shen[1], Ning Chai[1], Bin Qian[1], Yongchao Cai[1], Wenya Wang[1], Ziyi Yin[1], Haifeng Zhang[1,2], Xiaobo Zheng[1,2], Zhengguang Zhang[1,2]*

[1]Department of Plant Pathology, College of Plant Protection, Nanjing Agricultural University, and Key Laboratory of Integrated Management of Crop Diseases and Pests, Ministry of Education, Nanjing, China; [2]The Key Laboratory of Plant Immunity, Nanjing Agricultural University, Nanjing, China

**Abstract** The production of reactive oxygen species (ROS) is a ubiquitous defense response in plants. Adapted pathogens evolved mechanisms to counteract the deleterious effects of host-derived ROS and promote infection. How plant pathogens regulate this elaborate response against ROS burst remains unclear. Using the rice blast fungus *Magnaporthe oryzae*, we uncovered a self-balancing circuit controlling response to ROS in planta and virulence. During infection, ROS induces phosphorylation of the high osmolarity glycerol pathway kinase MoOsm1 and its nuclear translocation. There, MoOsm1 phosphorylates transcription factor MoAtf1 and dissociates MoAtf1-MoTup1 complex. This releases MoTup1-mediated transcriptional repression on oxidoreduction-pathway genes and activates the transcription of MoPtp1/2 protein phosphatases. In turn, MoPtp1/2 dephosphorylate MoOsm1, restoring the circuit to its initial state. Balanced interactions among proteins centered on MoOsm1 provide a means to counter host-derived ROS. Our findings thereby reveal new insights into how *M. oryzae* utilizes a phosphor-regulatory circuitry to face plant immunity during infection.

*For correspondence:
zhgzhang@njau.edu.cn

Competing interests: The authors declare that no competing interests exist.

## Introduction

During co-evolution with pathogens, plants have developed an innate immune system by sensing pathogen-associated molecular patterns (PAMPs), such as flagellin and chitin produced, respectively, by bacteria and fungi (*Jones and Dangl, 2006*; *Liu et al., 2014*). Upon recognition, pattern-recognition receptors (PRRs) transduce the signals to the downstream components that initiate the immune response (*Boutrot and Zipfel, 2017*; *Zipfel, 2014*), which was referred to as PAMP-triggered immunity (PTI) (*Boutrot and Zipfel, 2017*; *Jones and Dangl, 2006*). This PRR to PTI processes often involves components of MAP kinase signaling pathways, callose deposition, ROS burst, and pathogenesis related (PR) gene expression.

The NADPH oxidase-mediated production of ROS is one of the earliest PTI responses restricting pathogen invasion. In *Arabidopsis thaliana*, phosphorylated BIK1 activates the NADPH oxidase RBOHD through protein phosphorylation to trigger ROS burst. In rice (*Oryza sativa*), the LysM domain of the chitin-elicitor binding protein (CEBiP) binds to chitin and associates with the bifunctional plant receptor OsCERK1 that phosphorylates the downstream receptor-like cytoplasmic kinase 185 (OsRLCK185) for the activation of PTI by causing the ROS burst (*Yamaguchi et al., 2013*). In

addition, Rac1, the Rac/ROP small G-protein, which interacts with a defense-related NADPH oxidase RbohB that response to chitin elicitor for ROS production (*Nagano et al., 2016*).

During host infection, pathogens secrete numerous antimicrobial proteins, including superoxide dismutase, catalases, and peroxidases, to evade host immunity (*Kawasaki et al., 1997*; *Lanfranco et al., 2005*; *Molina and Kahmann, 2007*). In the corn smut fungus *Ustilago maydis*, the Protein Essential During Penetration 1 (Pep1) protein directly interferes with ROS generation by inhibiting peroxidase activities (*Hemetsberger et al., 2012*). In the tomato leaf mold fungus *Cladosporium fulvum*, the LysM domain-containing Effector Extracellular Protein 6 (Ecp6) circumvents chitin-induced immunity by sequestering host chitin oligomers (*de Jonge et al., 2010*). Similarly, *M. oryzae* secreted LysM Protein 1 (MoSlp1) competes with OsCEBiP for chitin binding, thereby preventing the activation of rice PTI (*Chen et al., 2014*; *Mentlak et al., 2012*). Recently, the rice tetratricopeptide repeat protein OsTPR1 was shown to interact with *M. oryzae* chitinase MoChia1 in the apoplast. In addition, the competitive binding of OsTPR1 by MoChia1 allows the accumulation of free chitin to reestablish the host immune response (*Yang et al., 2019*).

*M. oryzae* causes rice blast and is also a hemibiotrophic fungus in need of host nutrients for propagation (*Wilson and Talbot, 2009*; *Zhang et al., 2016*). How *M. oryzae* response to host-derived signals to circumvent plant immunity during infection remain a much-debated question. In *M. oryzae* and other fungal pathogens, G-protein/cAMP signaling plays an important role in the perception of host surface cues (*Choi and Dean, 1997*; *Liu et al., 2007*). The non-canonical G-protein coupled receptor (GPCR) Pth11 that functions upstream of G-protein/cAMP signaling is also important for surface perception in *M. oryzae* (*DeZwaan et al., 1999*; *Kou et al., 2017*). A previous study identified that the sensor kinase protein MoSln1 functions to sense glycerol and facilitates host penetration of *M. oryzae* (*Ryder et al., 2019*). In spite of any ROS receptor remaining to be identified, *M. oryzae* is known to contain several conserved MAP kinase pathways, including MoMst11-MoMst7-MoPmk1, MoMck1-MoMkk1-MoMps1, and MoSsk2-MoPbs2-MoOsm1 in conferring signal transduction during infection (*Yin et al., 2016*; *Zhang et al., 2016*). Among them, the Hog1 homolog, MoOsm1, which mediated the osmoregulation pathway is essential for the response to hyperosmotic stress through transcription factor MoMsn2 (*Dixon et al., 1999*; *Zhang et al., 2014*). Additional studies also found that the osmoregulation pathway is important for the response to oxidative species and resistance to fungicides (*Kim et al., 2009*).

Previous studies demonstrated that the bZIP transcription factor MoAp1 is important in response to oxidative stress by activating a suite of antioxidant genes during ROS stress (*Guo et al., 2011*), and MoAtf1 is also important in response to hyperosmotic stress and ROS stress (*Guo et al., 2010*). The ΔMoatf1 mutant was hypersensitive to oxidative stress, exhibited the reduced expression of several extracellular peroxidase and laccase genes, and failed to suppress the accumulation of ROS around the infection sites (*Guo et al., 2010*). To understand how *M. oryzae* responds to the ROS-mediated stress and triggered the downstream signaling pathway for ROS tolerance, we sought upstream to identify kinase which regulates MoAtf1 in response to ROS stress. We found that host-derived ROS induces the MoOsm1-mediated MAPK pathway to activate MoAtf1 phosphorylation. In addition, we identified phosphorylated MoAtf1 initiates the transcription of *MoPTP1/MoPTP2* under ROS stress which function on the dephosphorylation of MoOsm1. The process of MoOsm1/MoPtps-mediated phosphor-regulatory feedback loop function as a switch which not only enhanced virulence of *M. oryzae* under ROS stress but also control the virulence that keep the rice cells alive during hemibiotrophic growth.

## Results

### *M. oryzae* infection induces ROS accumulation in rice

During *M. oryzae* infection, the pathogen and rice interaction results in either disease or host immunity. In infection of rice cultivar LTH by *M. oryzae* wild-type strain Guy11, the sequences of various developmental stages are as follows: primary hyphae to appressorium differentiation (<20 hpi), the penetration of epidermis (20 hpi), formation of the bulbous infection hyphae (IH) inside the host cell (24 hpi), and spreading into the neighboring cells (36 hpi) for further infection (48–72 hpi). In a moderate resistance cultivar-strain interaction, such as between K23 and Guy11, few and restricted lesions were present (*Liu et al., 2018*; *Yin et al., 2020*). The hyphae grew poorly in the leaf-sheath

cells (24 and 36 hpi) and were restricted to the primary infected cells at 48 hpi until eventual spreading into adjacent cells (60 and 72 hpi) (*Figure 1A*).

We used DAB staining to estimate ROS accumulation in response to *M. oryzae* infection. Rice cultivar K23 infected with Guy11 yielded reddish-brown precipitates around the appressoria and infected hypha at 24, 36, 48, and 60 hpi. Over 40% of the infected cells were stained brown at 24 hpi and/or 36 hpi. When observation was made at 60 hpi, nearly 10% of infected cells were still filled with ROS. In contrast, the accumulation of ROS was barely detectable in the susceptible LTH cultivar infected by Guy11. The rate of cells stained with DAB was no more than 20% at 24, 36, and 48 hpi (*Figure 1B and C*). These results indicated that rice elaborates ROS as a barrier to infection as early as 24 hpi, and scavenging of host ROS may be necessary for further expansion of *M. oryzae* during infection.

## MoOsm1 phosphorylation in response to oxidative stress

MoOsm1 is an essential component of the osmoregulation pathway, and a previous study indicated that the deletion of the *MoOSM1* gene resulted in hypersensitivity to oxidative stress (*Dixon et al., 1999*). To understand this MoOsm1-mediated ROS response, we constructed the strain expressing MoOsm1-GFP, in which the expression of the C-terminal GFP fusion protein is under the control of the native *MoOSM1* promoter, and we tested phosphorylation of MoOsm1 using rice seedlings of both compatible pair (Guy11 and LTH) and the relative resistance pair (K23 and LTH). Total proteins were extracted at 0, 8, 20, 24, 48, and 72 hpi and the proteins bound to the anti-GFP beads were eluted and analyzed by anti-p38 MAPK (Figureure 2A and 2B). The results showed that the phosphorylation of MoOsm1 reached high levels at 24 and 48 hpi before dropping at 72 hpi in the K23 and Guy11 pair (Figureure 2B), but not in the LTH and Guy11 pair (Figureure 2A). Given the time of 24 and 48 hpi correlates with ROS levels, we speculated that ROS burst might have a role in MoOsm1 phosphorylation. When K23 was treated with 0.5 µM diphenyleneiodonium (DPI) that inhibits the activity of plant NADPH oxidases and thereby ROS (*Bolwell et al., 1998*; *Grant et al., 2000*; *Zhang et al., 2009*), MoOsm1 phosphorylation was significantly reduced at 24 hpi and 36 hpi (Figureure 2C). OsRbohA, an important NADPH oxidase, is critical for the ROS generation in rice, and OsRbohA-overexpressing transgenic plants exhibited higher ROS production (*Wang et al., 2016*). To further understand the function of MoOsm1 in response to host-produced ROS, we detected the phosphorylation level of MoOsm1 on the OsRbohA-ox line during the infection. The results showed that the phosphorylation of MoOsm1 was induced in the OsRbohA-ox line compared with the NPB lines (*Figure 2D*). To confirm that ROS levels were correlated with MoOsm1 phosphorylation, an in vitro assay was carried out, showing a similar result. When Guy11 was treated with 5 mM $H_2O_2$ for 0, 10, 30, and 60 min, an enhanced phosphorylated MoOsm1 was observed at 10 min (*Figure 2—figure supplement 1A*).

In addition, we observed the localization of MoOsm1 in response to oxidative stress. Under normal conditions, MoOsm1 was equally present in both the cytoplasm and the nucleus in conidium and mycelium. Following treatment with 5 mM $H_2O_2$ for 10 min, an enhanced nuclear localization pattern was observed in conidia (76.32 ± 17.83%) and hypha (67.48 ± 19.33) (*Figure 2E* and S1B). We also fused a nuclear export signal (NES) sequence to MoOsm1 and found that 5 mM $H_2O_2$ did not exhibit any significant effect to the location of MoOsm1$^{NES}$-GFP (Figure S1C). When $H_2O_2$ treatment was extended to 30 min, the localization of MoOsm1 recovered to the default distribution pattern (*Figure 2E*, *Figure 2—figure supplement 1B*). When a red fluorescent protein (RFP) was fused to histone H1 marking the nucleus, an enrichment of MoOsm1 in the nucleus following $H_2O_2$ treatment for 10 min was clearly visualized (*Figure 2F*). We also performed western blotting analysis using extracted nuclear proteins and found MoOsm1-GFP was significantly enriched in the nucleus compared with wild type upon 10 min $H_2O_2$ treatment (*Figure 2G*). These results suggested that MoOsm1 responds to oxidative stress by accumulating in the nucleus.

## MoOsm1 undergoes dimer to monomer transition under oxidative stress-induced phosphorylation

As $H_2O_2$ induces phosphorylation and nuclear localization of MoOsm1, we hypothesized that the phosphorylation is relevant to its localization. To test this, we first identified the oxidation stress-dependent phosphorylation site of MoOsm1. We purified the MoOsm1-GFP protein from the

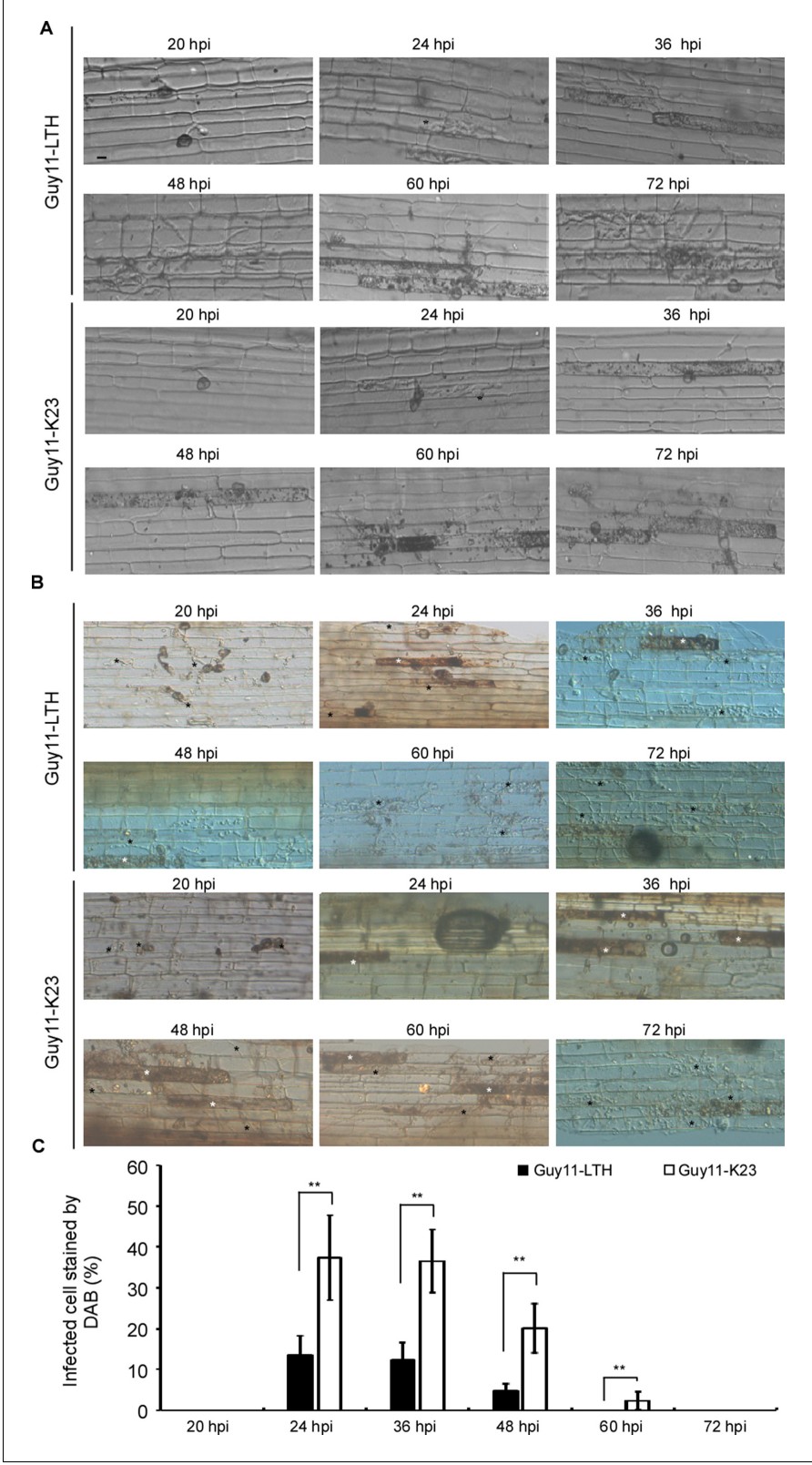

**Figure 1.** Time-course images of ROS accumulation during rice sheath infection by *M. oryzae*. (**A**) The conidial suspension of Guy11 ($1 \times 10^5$ spores/ml) was inoculated in the excised rice sheath of 4-weeks-old rice seedlings LTH and K23. The invasive hyphae growth was observed at 20, 24, 36, 48, 60, and 72 hpi. Black asterisks represent the bulbous infection hyphae (IH). (**B**) DAB staining shows ROS accumulation in rice LTH and K23 cells at various

*Figure 1 continued on next page*

*Figure 1 continued*

time points following infection. Black asterisks represent infected cells without ROS and white asterisks represent cells stained by DAB. (C) Infected cells stained by DAB. Over 50 infected rice cells were calculated with three replicates each time. '%" represents to the rate of infected cells which stained by DAB in all infected cells. Three independent biological experiments were performed and yielded similar results. Error bars represent standard deviation, and asterisks represent significant differences between the different strains (p<0.01).

$\Delta Moosm1/MoOSM1$-GFP strain that was treated with $H_2O_2$ for 10 min and found that threonine (T) 171 and tyrosine (Y) 173 were the corresponding phosphorylation sites (*Figure 3—figure supplement 1A*) through mass spectrometry analysis. We then generated a constitutively activated phosphomimetic mutation of MoOsm1$^{Y173D}$ and an inactivated mutation of MoOsm1$^{Y173A}$ as a validation step. MoOsm1-GFP, MoOsm1$^{Y173D}$-GFP, and MoOsm1$^{Y173A}$-GFP were expressed in the $\Delta Moosm1$ mutant, and the localization was observed using a fluorescence microscope. The results showed that the phosphomimetic mutation of MoOsm1$^{Y173D}$ had more nuclear accumulation relative to MoOsm1 and MoOsm1$^{Y173A}$ (*Figures 3A, B and C*). These findings indicated that the phosphorylation of MoOsm1 tyrosine 173 is important for its nuclear accumulation.

As a previous study suggested that the p38 MAPK kinase forms dimers with swapped activation segments (*Rothweiler et al., 2011*), suggesting that MoOsm1 may also undergo changes in dimerization. To test this via a co-immunoprecipitation (co-IP) approach, we co-introduced the *MoOSM1*-FLAG, *MoOSM1*-GFP, and the point-mutation constructs into the protoplasts of Guy11. Total proteins were extracted from the transformants, and MoOsm1 was detected using the anti-FLAG and anti-GFP antibodies. In proteins eluted from MoOsm1 and MoOsm1 $^{Y173A}$ anti-GFP beads, MoOsm1-FLAG was also detected. However, when co-introduced *MoOSM1* $^{Y173D}$-FLAG and *MoOSM1*$^{Y173D}$-GFP, the interaction was not found (*Figure 3D*). In addition, the interaction between MoOsm1 and MoOsm1$^{T171D}$ and the localization of MoOsm1$^{T171D}$ was also detected (*Figure 3—figure supplement 1B and C*). Collectively, the results suggested that phosphorylation of tyrosine 173 but not threonine 171 inhibits interaction. Using the native-PAGE analysis, we found that MoOsm1 is present in the form of both monomers and dimers, while only monomers were detected in the MoOsm1$^{Y173D}$ strains (*Figure 3E*). We also expressed the His-MoOsm1 protein in vitro and purified it by AKTA pure (GE healthcare) with gel-filtration chromatography. We separated four putative peaks for further verification by western blot. The results showed that only peaks I and II were identified at 110 kd and 55 kd, suggesting that MoOsm1 form dimer (*Figure 3F*). The dimerization was further verified by the bimolecular fluorescence complementation (BiFC) assay. cYFP-*MoOSM1* and *MoOSM1*-nYFP pair, cYFP-*MoOSM1*$^{Y173A}$ and *MoOSM1*$^{Y173A}$ -nYFP pair fusion constructs were co-introduced into Guy11 protoplasts and transformants obtained. The recombined YFP fluorescence signal was detected in the cytoplasm containing the corresponding protein pairs (*Figure 3G*). Moreover, upon $H_2O_2$ treatment for 10 and 15 min, the YFP signal of cYFP-*MoOSM1* and *MoOSM1*-nYFP pair was reduced in the nucleus, in contrast to cYFP-*MoOSM1* $^{Y173A}$ and *MoOSM1*$^{Y173A}$-nYFP pair that showed the default localization pattern, suggesting that the monomeric form of MoOsm1 is involved in the nuclear localization under the oxidative stress (*Figure 3G*).

Phosphorylation of MoOsm1 was further evaluated using Phos-tag gel electrophoresis. Total extracts were treated with either phosphatase or phosphatase inhibitor (PI), and the mobility shift was examined by immunoblotting proteins with the anti-GFP antibody. The induced MoOsm1 mobility shift was found in the phosphatase treated wild-type cells, but not in the PI-treated cells. A similar band shift was observed in the extracts from the cytoplasm with the phosphatase treated strain. The decreased mobility of MoOsm1-GFP purified from the nucleus was exhibited compared to the phosphatase treated strain, indicating a higher level of MoOsm1 phosphorylation in the nucleus (*Figure 3H*). These results suggested that MoOsm1 could be phosphorylated and transferred into the nucleus.

## MoOsm1 phosphorylates MoAtf1 in vivo and in vitro

The MAPK kinase signaling pathways regulate developmental processes by targeting various downstream transcription factors or target genes. Several putative transcription factors were proposed to function downstream of MoOsm1, including MoAtf1, MoAp1, and MoMsn2 (*Li et al., 2012*; *Zhang et al., 2014*). To understand the function of MoOsm1 phosphorylation and translocation, we

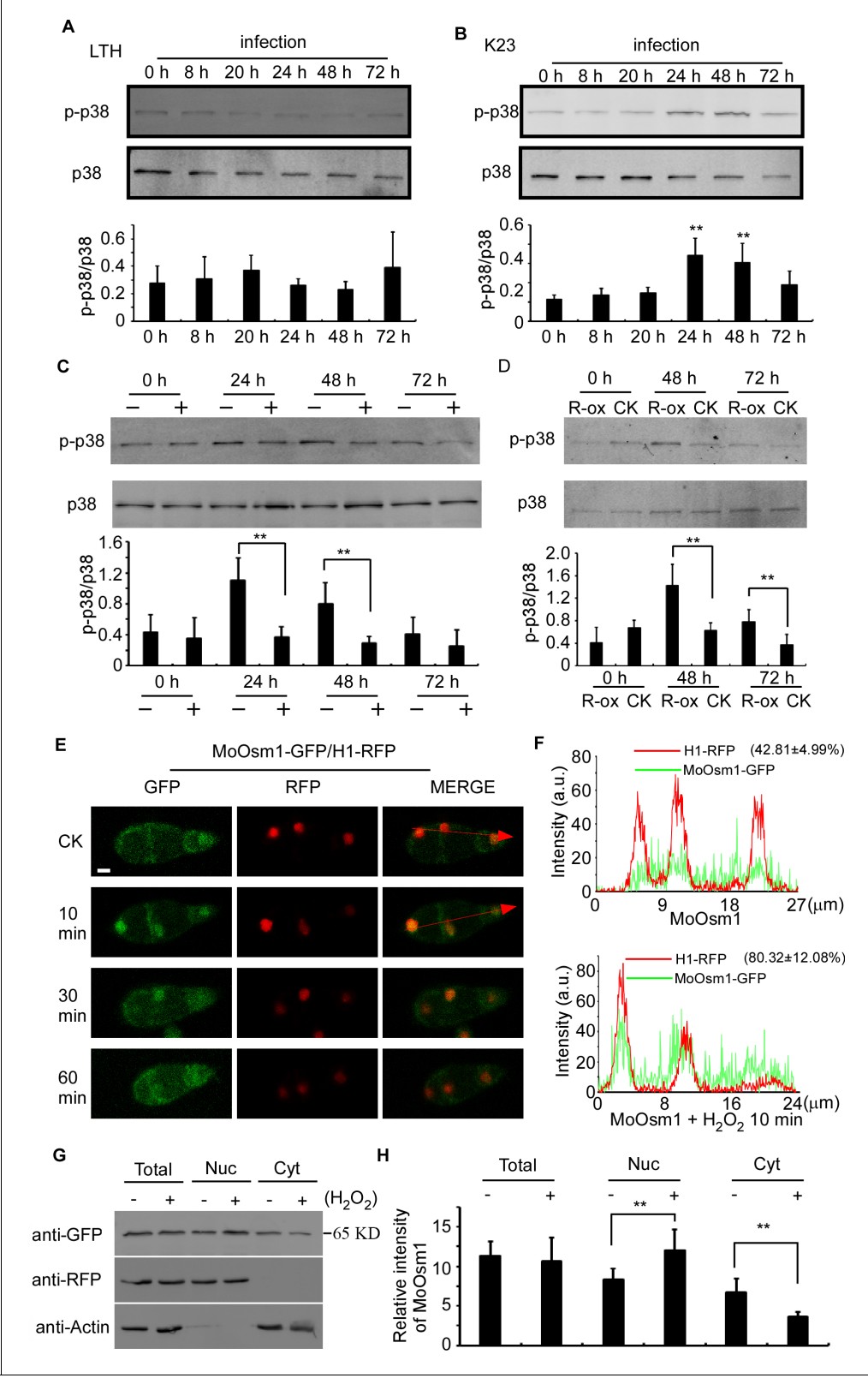

**Figure 2.** MoOsm1 phosphorylation and nuclear translocation in response to ROS stress. (**A**) Phosphorylation of MoOsm1 in Guy11 in infection of LTH. Total proteins were extracted from LTH leaves 0, 8, 20, 24, 48, and 72 hr. Eluted proteins bound to the anti-GFP beads were analyzed by the antiphospho-p38 antibody, with the p38 antibody used as a control. The extent of phosphorylation was estimated by calculating the amount of antiphospho-p38 compared to the p38 (the histogram underneath the blot). Error bars represent standard deviation. (**B**) Phosphorylation of MoOsm1 in

*Figure 2 continued on next page*

*Figure 2 continued*

Guy11 in infection of K23. MoOsm1 phosphorylation was induced at 24 and 48 hpi. Error bars represent standard deviation, and asterisks represent significant differences between the different strains. Values are the means of 3 replications, and error bars represent the SD (n = 3). The asterisks indicate a significant difference (Duncan's new multiple range test, p<0.01). (**C**) DPI treatment in rice decreases the phosphorylation levels of MoOsm1 during infection. Total proteins were extracted from K23 leaves 0, 24, 48, and 72 hr with (+) or without (−) DPI treatment. MoOsm1 was purified and analyzed by the antiphospho-p38 antibody, with the p38 antibody used as a control. (**D**) Overexpression of OsRbohA induced the phosphorylation of MoOsm1. Total proteins were extracted from OsRbohA-ox leaves at 0, 48 and 72 hpi. MoOsm1 was purified and analyzed by the antiphospho-p38 antibody. NPB was used as the CK. R-ox represents OsRbohA-ox lines. (**E**) Localization of MoOsm1 under oxidative stress. Fluorescence observation of conidia treated with $H_2O_2$ for 10, 30, and 60 min. MoOsm1-GFP and H1-RFP were observed by confocal fluorescence microscopy. Bars = 5 μm. (**F**) Fluorescence intensity of MoOsm1-GFP/H1-RFP was observed with 10 min $H_2O_2$ treatment and CK. The green line represents MoOsm1-GFP while the red line represents H1-RFP. Insets highlight areas analyzed by line-scan. The number represents the quantification of GFP and RFP signals by ImageJ. (**G**) An equal amount ($8 \times 10^6$ spore/ml x 60 ml) of conidia (with 10 min $H_2O_2$ treatment or not) were divided into three parts for extraction of the total, nuclear, and cytoplasm proteins. Equal amounts of the total, nuclear and cytoplasm proteins were separated by SDS-PAGE, and MoOsm1 was detected by western blotting using the anti-GFP antibody. Bands of MoOsm1-GFP were detected at 65kD. The intensity of western blotting bands was quantified with the ODYSSEY infrared imaging system (application software Version 2.1). The intensity of MoOsm1 was compared between the conidia without treatment (-) and conidia with 10 min of $H_2O_2$ treatment (+) among total proteins, nuclear proteins, and cytoplasmic proteins. H1 (a nucleus marker) and actin (a cytoplasm marker) were detected by western blotting analysis. Bars denote standard errors from three independent experiments. Asterisks indicate significant differences (Duncan's new multiple range test p<0.01).

The online version of this article includes the following figure supplement(s) for figure 2:

**Figure supplement 1.** Localization of MoOsm1 under $H_2O_2$ stress in the conidium and mycelium.

first validated the interaction between MoOsm1 and MoAtf1 by co-IP. The MoAtf1-FLAG, MoOsm1-GFP, MoOsm1Y173D-GFP, MoOsm1Y173A-GFP, and MoOsm1NES-GFP fusion constructs were introduced into the protoplasts of Guy11, and proteins were extracted from the transformants. MoAtf1 and MoOsm1 were detected using the anti-FLAG and anti-GFP antibodies. In proteins eluted from anti-GFP beads, MoAtf1 was detected in the elution among MoOsm1, MoOsm1Y173D, and MoOsm1Y173A, but not MoOsm1NES (*Figure 4A*), indicating that MoOsm1 interacts with MoAtf1. The interaction was further confirmed by the BiFC assay. The recombinant YFP fluorescence signal was detected among MoOsm1-MoAtf1, MoOsm1Y173D-MoAtf1, and MoOsm1Y173A-MoAtf1 in the nucleus in comparison with MoOsm1NES (*Figure 4B*), suggesting that nuclear localization is important for MoOsm1 and MoAtf1 interaction.

Given that the phosphorylated MoOsm1 is translocated into the nucleus under ROS stress and interacts with MoAtf1, MoOsm1 could phosphorylate MoAtf1. To test this hypothesis, we generated a *MoATF1-GFP* construct and introduced it into both Guy11 and the Δ*Moosm1* mutant strain and then analyzed MoAtf1-GFP using Phos-tag gel electrophoresis. Total extracts were treated with either a phosphatase or a PI, and the mobility shift was examined by immunoblotting proteins with the anti-GFP antibody. The induced MoAtf1 mobility shift was observed in the phosphatase treated wild-type cells but not in the untreated or PI-treated cells. A similar band shift was not observed in the extracts from the untreated Δ*Moosm1* mutant. The increased mobility of MoAtf1-GFP in the untreated Δ*Moosm1* mutant compared to the untreated wild-type strain indicated a higher level of MoAtf1 phosphorylation in the wild-type strain (*Figure 4C*). These results suggested that MoOsm1 regulates MoAtf1 through protein phosphorylation.

When observing MoAtf1 localization, we found that its nuclear localization remains unchanged in the Δ*Moosm1* mutant, indicating that MoOsm1-mediated MoAtf1 phosphorylation does not seem to have an effect on its localization (*Figure 4D*). To understand the underlying mechanism, we tested whether any serine/threonine located in the front of the nuclear localization signal was involved. We generated the point-mutation mutants (S29A, S117A, S124A, S152A, T161A, S308A, and S334A) corresponding to six S (Ser) and one T (Thr) residues in MoAtf1 (*Figure 4E*). In vivo phosphorylation analysis showed that the phosphorylation level of MoAtf1 was dampened in the S124A mutation strain but not in other mutants (*Figure 4F*). In addition, MoAtf1-GST, the constitutively unphosphorylated MoAtf1S124A-GST, and MoOsm1-His fusion proteins were also obtained and analyzed. The results showed a relatively low level of MoAtf1S124A phosphorylation (*Figure 4—figure supplement 1*). These results collectively suggested that MoOsm1 phosphorylates MoAtf1 on serine 124.

Since MoAtf1 contributes to the full virulence of *M. oryzae*, we questioned if this phosphorylation on serine 124 could also regulate the virulence of *M. oryzae*. We then introduced the constitutively

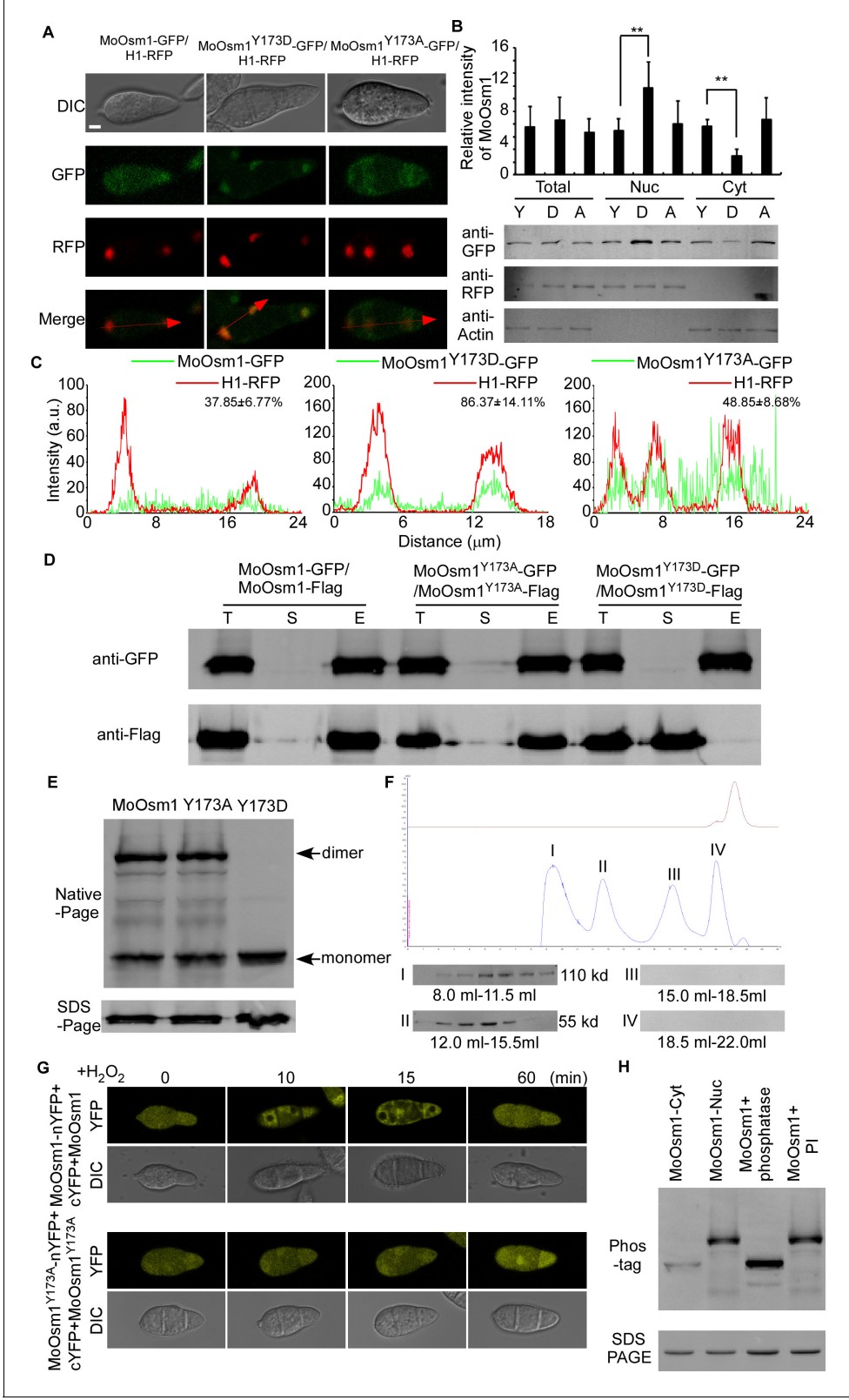

**Figure 3.** Phosphorylation-mediated monomerization of MoOsm1 results in its nuclear accumulation. (**A**) Localization of MoOsm1, MoOsm1^Y173D, and MoOsm1^Y173A in conidia. MoOsm1-GFP, MoOsm1^Y173D-GFP, MoOsm1^Y173A-GFP, and H1-RFP were observed by confocal fluorescence microscopy. Bars = 5 µm. (**B**) Nuclear/cytoplasmic proteins harvested from conidia of Δ*Moosm1/MoOsm1-GFP*, Δ*Moosm1/MoOsm1^Y173D-GFP*, and Δ*Moosm1/MoOsm1^Y173A-GFP* strains were separated by SDS-PAGE. The intensity of MoOsm1 was compared among total proteins, nuclear proteins, and

*Figure 3 continued on next page*

Figure 3 continued

cytoplasmic proteins. 'Y' represents MoOsm1, 'D' represents MoOsm1$^{Y173D}$, and 'A' represents MoOsm1$^{Y173A}$. Bars denote standard errors from three independent experiments. Asterisks indicate significant differences (Duncan's new multiple range test p<0.01). (C) Fluorescence intensity of MoOsm1-GFP/H1-RFP in ΔMoosm1/MoOsm1-GFP, ΔMoosm1/MoOsm1$^{Y173A}$-GFP, and ΔMoosm1/MoOsm1$^{Y173D}$-GFP strains. The number represents the quantification of GFP and RFP signals by ImageJ. (D) Co-IP assay. Western blot analysis of total proteins (T) extracted from various transformants, suspension proteins (S), and elution proteins (E) eluted from anti-GFP beads. MoOsm1-GFP, MoOsm1$^{Y173D}$-GFP, MoOsm1$^{Y173A}$-GFP, and MoOsm1-FLAG were detected with respective antibodies. (E) Immunoblot analysis of proteins extracted from MoOsm1-GFP/ΔMoosm1, MoOsm1$^{Y173A}$-GFP/Δ Moosm1, and MoOsm1$^{Y173D}$-GFP/ΔMoosm1 strains. (F) Gel-filtration chromatography assay of MoOsm1 dimerization. MoOsm1 was separated by the AKTA protein purification system (GE healthcare). Four protein peaks (I, II, III, and IV) were detected by western blot analysis. The brown line represents the ion peak. (G) BiFC assays for MoOsm1-nYFP and cYFP-MoOsm1 interactions. Conidia were treated with H$_2$O$_2$. MoOsm1$^{Y173A}$-nYFP and cYFP-MoOsm1$^{Y173A}$ were used as control. (H) In vivo phosphorylation analysis of MoOsm1. Nuclear and cytoplasmic MoOsm1 phosphorylation was analyzed by Mn$^{2+}$-Phos-tag SDS-PAGE and normal SDS-PAGE, respectively. Total proteins treated with alkaline phosphatase (phosphatase) and phosphatase inhibitor (inhibitor) were used as control.

The online version of this article includes the following figure supplement(s) for figure 3:

**Figure supplement 1.** Thr171 and Tyr173 are important phosphorylation sites in MoOsm1.

---

expressed MoAtf1$^{S124D}$ and MoAtf1$^{S124A}$ into ΔMoatf1, respectively. Virulence testing showed no differences between wild type (Guy11) and MoAtf1$^{S124D}$ expression strains on LTH cultivar, while both ΔMoatf1 and ΔMoatf1/MoAtf1$^{S124A}$ mutants exhibited restricted lesions. On the K23 cultivar, however, the wild-type strain caused few typical lesions (gray spots with brown margins), in contrast to the MoAtf1$^{S124D}$ expression strains that caused even more lesions and also larger lesion areas (*Figure 4G and H*). This result demonstrated that phosphorylation of MoAtf1 on residue 124 is important for the virulence of *M. oryzae* on rice.

## MoAtf1 phosphorylation disrupts its interaction with MoTup1 that affects the expression of oxidoreduction- pathway factors

During screening for MoAtf1-interacting proteins, we identified MoTup1, a previously characterized conserved transcription repressor (*Figure 5—figure supplement 1*; *Chen et al., 2015*). We found that MoAtf1 interacts with MoTup1 in yeast two-hybrid (Y2H), pull down and co-IP assay (*Figure 5— figure supplement 2A, B and C*). We then constructed the MoAtf1$^{S124D}$ and MoAtf1$^{S124A}$ alleles to verify whether the phosphorylation of MoAtf1 abolishes its interaction with MoTup1. Indeed, compared with the MoAtf1 and MoAtf1$^{S124A}$, MoAtf1$^{S124D}$ could not interact with MoTup1 (*Figure 5— figure supplement 2A, B and C*), indicating that the phosphorylation of MoAtf1 on S124 causes its disassociation from MoTup1.

We have also performed a genome-wide ChIP-Seq assay to evaluate whether MoAtf1 has a role in regulating any pathogenicity-related genes. We found that MoAtf1 binds to the upstream regions of 574 open reading frames (ORFs) (*Supplementary file 1*). Among them, 16 were involved in the oxidation-reduction process, including six containing the signal peptide (*Figure 5*). We then validated ChIP-Seq findings by electrophoretic mobility shift assays and qRT-PCR. The results showed that these genes were down-regulated in the ΔMoatf1 mutant (*Figure 5* and *Figure 5—figure supplement 3A*) indicated that MoAtf1 positively regulates the oxidoreduction pathway.

Since MoAtf1 interacts with MoTup1, it is hypothesized that disruption of the MoAtf1-MoTup1 complex would reverse the suppression of genes involved in the oxidoreduction pathway. The phosphorylation at S124 of MoAtf1 dissociates the interaction between MoAtf1 and MoTup1 (*Figure 5— figure supplement 2*). The results showed that 13 out of 16 genes were upregulated after 10 min H$_2$O$_2$ treatment in Guy11, and 12 of 16 genes were upregulated at 24 hpi during infection (*Figure 5—figure supplement 3B and C*). The further qRT-PCR assay showed that H$_2$O$_2$ responsive genes were also upregulated in the ΔMoatf1/MoAtf1$^{S124D}$ strain (*Figure 5*). Collectively, the results showed that host-derived ROS induces the phosphorylation of MoOsm1 (*Figure 2*), which in turn phosphorylates MoAtf1 and disengages the MoAtf1-MoTup1 complex leading to the transcriptional activation of MoAtf1-regulating genes.

## MoPtp1/2 involved in the dephosphorylation of MoOsm1

As a hemibiotrophic fungus, *M. oryzae* needs to maintain host cells alive during the biotrophic stage. We found that the pathogen utilizes MoOsm1 phosphorylated MoAtf1 and, in turn, phosphorylated

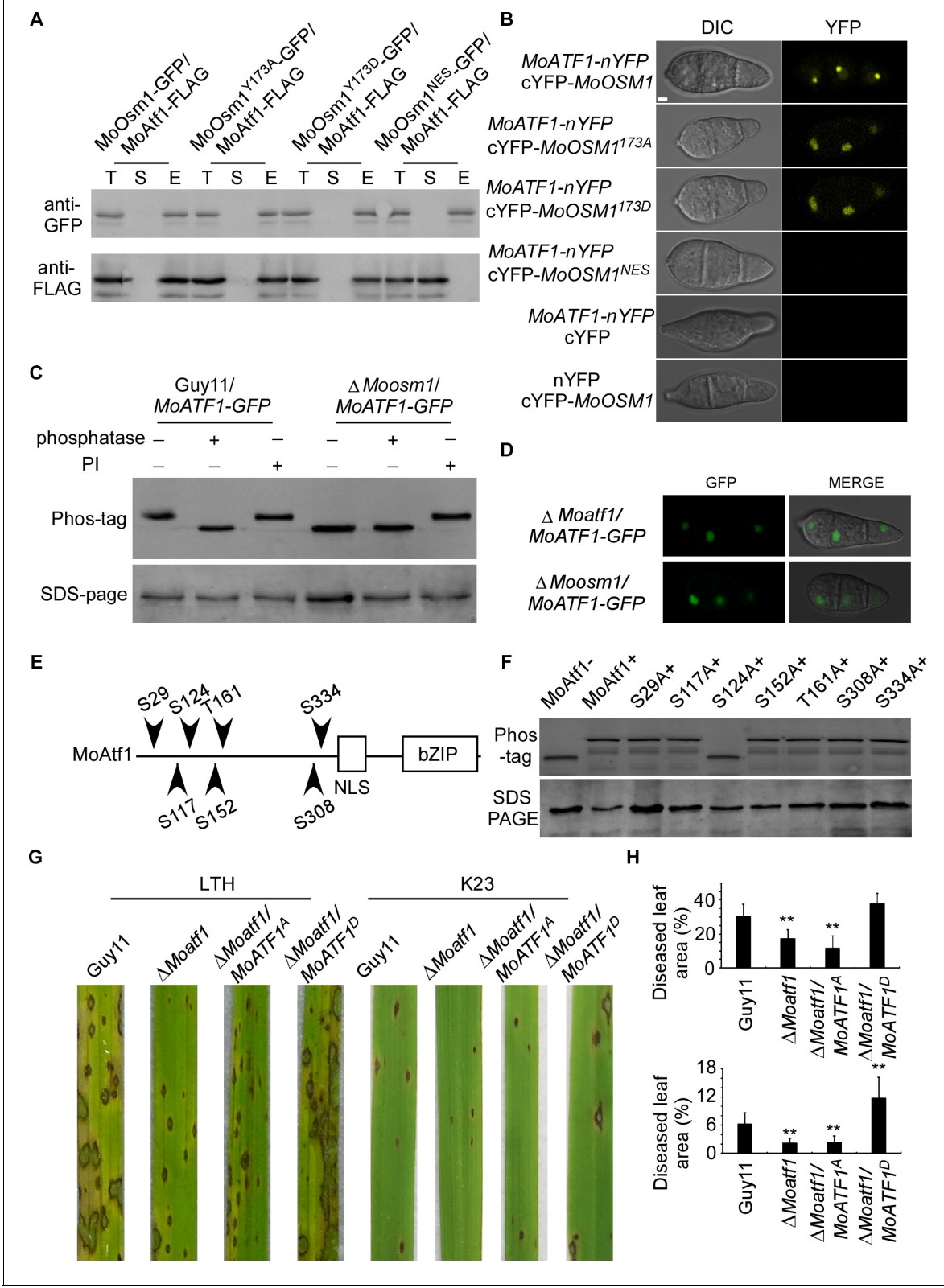

**Figure 4.** MoOsm1 phosphorylates MoAtf1 in the nucleus. (**A**) Co-IP assay. Western blot analysis of total proteins (**T**) extracted from MoOsm1-GFP/ MoAtf1-FLAG, MoOsm1$^{Y173A}$-GFP/MoAtf1-FLAG, MoOsm1$^{Y173D}$-GFP/MoAtf1-FLAG, MoOsm1$^{NES}$-GFP/MoAtf1-FLAG strains, suspension proteins (**S**), and elution proteins (**E**) eluted from anti-GFP beads. The presence of MoAtf1 and MoOsm1 was detected with the anti-GFP and anti-FLAG antibodies, respectively. MoOsm1$^{NES}$-GFP was used as a negative control. (**B**) BiFC assays for the interaction between MoOsm1 and MoAtf1. Strains expressed

*Figure 4 continued on next page*

*Figure 4 continued*

MoAtf1-nYFP and empty cYFP, cYFP-MoOsm1 and empty nYFP, MoAtf1-nYFP and MoOsm1[NES]-GFP were used as negative controls. Bars = 5 µm. (C) Phosphorylation of MoAtf1 by MoOsm1. MoAtf1-GFP proteins treated with alkaline phosphatase and phosphatase inhibitors (PIs) were separated by Mn$^{2+}$-Phos-tag and normal SDS-PAGE, respectively, and detected by the GFP antibody. (D) The localization of MoAtf1 in Δ*Moosm1* and Δ*Moatf1* mutants. MoAtf1-GFP was introduced in the Δ*Moosm1* and Δ*Moatf1* mutants, and the localization was observed by confocal fluorescence microscopy. (E) A model of MoAtf1 and mutants' constructs. White bars indicate the conserved bZIP domain and nuclear localization signal (NLS) domain. The black triangle indicates putative serine and threonine sites and the numbers indicate amino acid positions. (F) Identification of the phosphorylation sites of MoAtf1 by MoOsm1. In vivo phosphorylation analysis of MoAtf1 and the inactivation mutants (S to A and T to A). MoAtf1-GFP proteins treated with alkaline phosphatase (MoAtf1-), PIs (MoAtf1+), and the phosphorylation inactivation mutants treated with PIs (S29A+, S117A+, S124A+, S152A+, T161A +, S308A+, and S334A+) were separated by Mn$^{2+}$-Phos-tag SDS-PAGE and normal SDS-PAGE, respectively. (G) Pathogenicity assay. Four milliliters of conidial suspension ($5 \times 10^4$ spores/ml) of each strain were used for spraying on LTH and K23 and photographed 5 d after inoculation. (H) Diseased leaf area analysis. Data are presented as a bar chart showing the percentage of lesion area analyzed by Image J. Error bars represent SD and asterisks represent significant differences ('**' represents p<0.01).

The online version of this article includes the following figure supplement(s) for figure 4:

**Figure supplement 1.** Phosphorylation of MoAtf1 in vitro.

MoAtf1 dissociated with MoTup1 in responding to host immunity. We speculated that MoOsm1 regulation might also involve functions of protein phosphatases. Based on the finding that the budding yeast *Saccharomyces cerevisiae* protein phosphatases Ptps are involved in the dephosphorylation of Hog1, a homolog of MoOsm1 (*Lee et al., 2014*; *Murakami et al., 2008*), we characterized whether the Ptp homologs MoPtp1 and MoPtp2 have a role in dephosphorylating MoOsm1.

We first generated and verified the respective Δ*Moptp1* and Δ*Moptp2* mutant strains (*Figure 6— figure supplement 1*). We showed that the deletion of *MoPTP1* and *MoPTP2* exhibited no defects in the vegetative growth (*Figure 6—figure supplement 2A and B*). Further microscopic observations showed that conidiation was significantly reduced in the Δ*Moptp2* mutants, but not in the Δ*Moptp1* mutants (*Figure 6—figure supplement 2C and D*). To examine the role of MoPtp1/2 in virulence, we inoculated conidial suspensions of wild type, Δ*Moptp1*, Δ*Moptp2,* and the complemented strains on the susceptible rice cultivar CO-39. The Δ*Moptp2* mutant caused small and restricted lesions, compared to the Δ*Moptp1* mutant that was fully virulent 7 days post-inoculation (dpi) (*Figure 6—figure supplement 3A and B*). Given the possibility that MoPtp1 and MoPtp2 may have redundant functions, independent Δ*Moptp1*Δ*Moptp2* double mutants were also constructed, and the result indicated that the double mutant was mostly similar to Δ*Moptp2* in phenotypes (*Figure 6—figure supplement 3A and B*).

We used Y2H and co-IP assay to examine the interaction between MoOsm1 and MoPtp1/2 (*Figure 6—figure supplement 3C and D*). To test whether MoPtp1/2 regulates the activity of MoOsm1 through protein dephosphorylation, Mn$^{2+}$-Phos-tag SDS-PAGE was performed. We found that the band of MoOsm1-GFP in both Δ*Moptp1* and Δ*Moptp2* mutants migrate as slow as that of the phosphorylated MoOsm1-GFP protein treated with the PI. MoOsm1-GFP in the wild-type strain displayed a migration pattern similar to proteins treated with the phosphatase (*Figure 6—figure supplement 3E*). These findings suggested that both MoPtp1 and MoPtp2 could dephosphorylate MoOsm1.

We further purified the MoOsm1-GFP protein from the Δ*Moptp2*/*MoOSM1-GFP* strain and found that tyrosine 173 was the phosphorylated (*Figure 6—figure supplement 4A*). So, we tested the phenotype of Δ*Moosm1*/*MoOsm1*[Y173D]and found that it had a similar defect as the Δ*Moptp2* mutant (*Figure 6—figure supplement 4B*), indicating that the constitutive phosphorylation of MoOsm1 results in virulence defect of *M. oryzae*.

## MoPtp2 plays a role in suppressing host ROS accumulation

To further understand the function of the Δ*Moptp1/2* mutants, we examined the appressorium formation and virulence. Appressorium formation of the Δ*Moptp1/2* mutants was normal (*Figure 6— figure supplement 5*). We then examined the penetration and invasive hyphal extension in rice sheath cells. After incubation with conidia suspension for 36 hr, more than 76% invasive hyphae of Guy11 were found to spread freely into adjacent cells, while the invasive Δ*Moptp2* hyphae (52%) and Δ*Moptp1*Δ*Moptp2* mutant (57%) were restricted in primary infected cells (*Figure 6A and B*). The Δ*Moptp1* mutant exhibited similar invasive hyphal extension as the wild type. All these results indicated that MoPtp2 plays an important role in infection and host colonization.

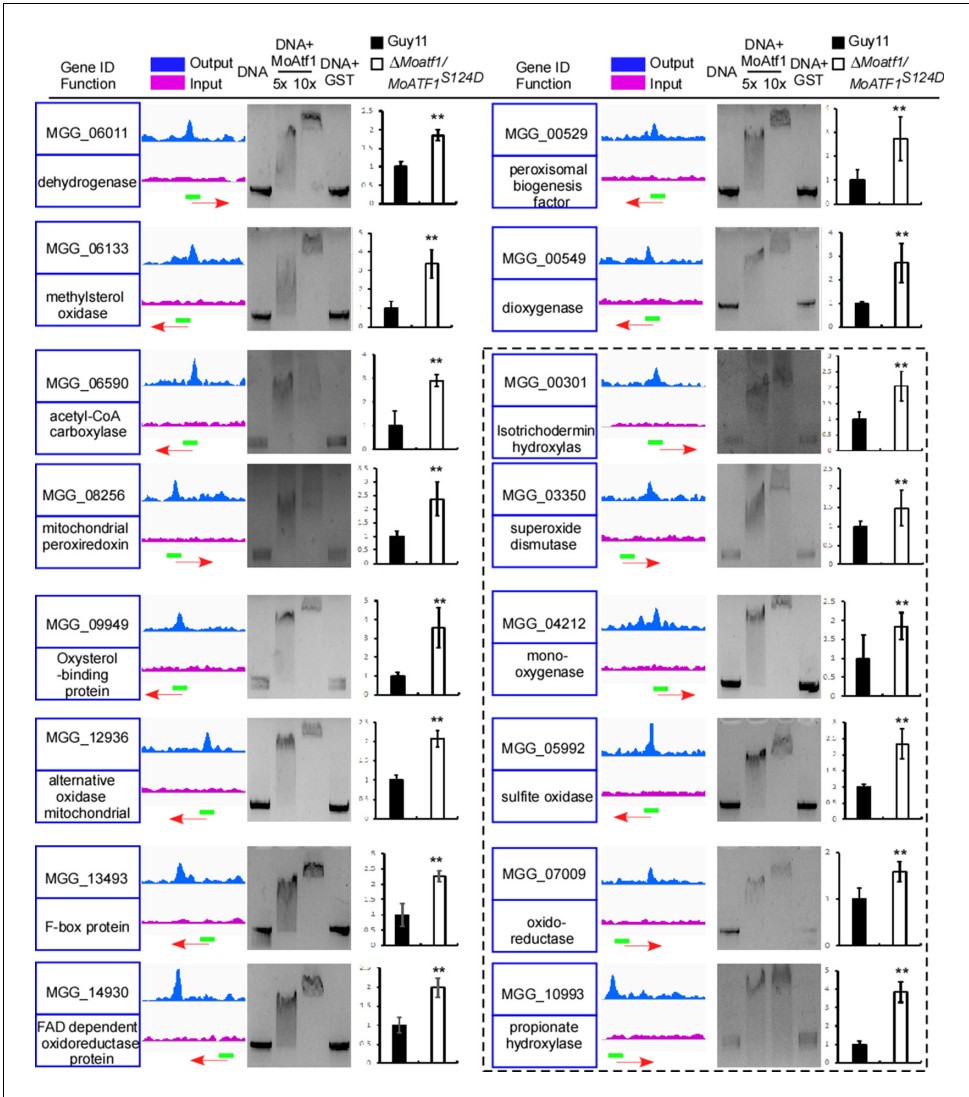

**Figure 5.** MoAtf1 is one of the regulators of the oxidation regulatory pathway. ChIP-Seq assays showed that MoAtf1 binds to the promoters of 16 oxidation regulation pathway genes. Output- and input-DNA was visualized in blue and pink. The red arrow indicates gene direction. The green bar represents the promoter region covered during ChIP-Sequencing. Blank in the black dashed line indicates genes encoding proteins containing signal peptides. The EMSA assay was performed to evaluate the relevant promoter of these genes binding with MoAtf1. The purified MoAtf1 was mixed with DNA, incubated for 20 min at 25˚C in binding buffer, and separated by 1.5% agarose gel. The qRT-PCR assay showed the expression of these genes in Guy11 and $\Delta Moatf1/MoAtf1^{S124D}$. Three independent biological experiments were performed, with three replicates each time that yielded similar results. Error bars represent standard deviation, and asterisks represent significant differences between the different strains (p<0.01).

The online version of this article includes the following figure supplement(s) for figure 5:

**Figure supplement 1.** Putative MoAtf1-interacting proteins identified by affinity purification.

**Figure supplement 2.** Phosphorylation of MoAtf1 on S124 blocks MoAtf1 and MoTup1 interaction.

**Figure supplement 3.** Expression levels of the oxidation regulatory pathway genes in the $\Delta Moatf1$ mutant.

As the deletion of *MoPTP2* caused defects in virulence, we speculated the increased accumulation of ROS around the infection sites accounts for the restricted growth. Using DAB staining, we found that the primary rice cells with the infectious hyphae of the $\Delta Moptp2$ mutant were stained intensely after incubation with conidia suspension for 36 hr. In contrast, cells with the Guy11 infectious hyphae were not stained by DAB (*Figure 6C*).

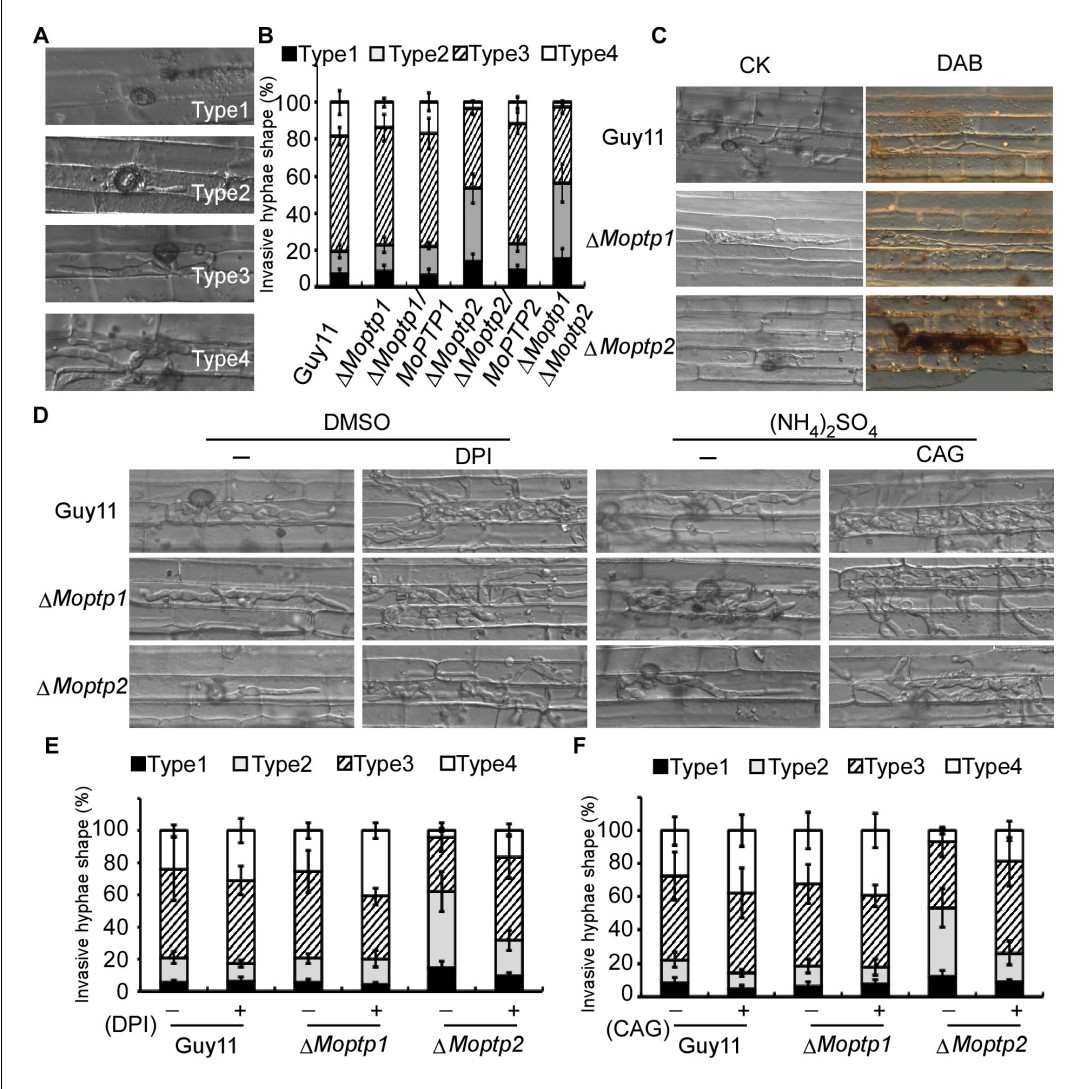

**Figure 6.** MoPtps are important in host-derived ROS scavenging by *M. oryzae*. (**A**) and (**B**) Excised rice sheaths from 3-week-old rice seedlings were inoculated with conidial suspension ($1 \times 10^5$ spores/ml). Infectious growth was observed at 24- and 36-hr post-inoculation (hpi). Appressorium penetration sites (n = 100) were observed and invasive hyphae (IH) were rated from type 1 to 4 (type1, no hyphal penetration with only appressoria formation; type2, IH with 1 or two short branches; type3, IH with at least three branches but the IH are short and extending within a plant cell; type 4, IH that has numerous branches and fully occupies the plant cell or even extended to an adjacent plant cell). The experiment was repeated three times. (**C**) DAB staining of the excised leaf sheath of infected rice 24 hpi. 50 infecting hyphae were counted per replicate and the experiment was repeated three times. (**D**) The excised sheath of rice was inoculated with conidial suspension after treated with or without 0.5 μM DPI dissolved in DMSO by three independent experiments. The rice sheath was also inoculated with the conidial suspension after treating with or without 0.2 U *Aspergillus niger* catalase (CAG, Sigma) dissolved in 10 mM $(NH4)_2SO_4$. Samples were harvested and observed 36 hr after inoculation. (**E**) and (**F**) Quantification of the infection progress with DPI and CAG treatment, over 50 infecting hyphae were counted per replicate, and the experiment was repeated three times. The online version of this article includes the following figure supplement(s) for figure 6:

**Figure supplement 1.** Southern blot analysis of *MoPTP1* and *MoPTP2* deletional mutants.
**Figure supplement 2.** MoPtps are not involved in mycelial growth and conidiation.
**Figure supplement 3.** MoPtp1 and MoPtp2 are involved in dephosphorylation of MoOsm1.
**Figure supplement 4.** Hyperphosphorylation on MoOsm1 Tyr173 causes virulence defect in *M. oryzae*.
**Figure supplement 5.** Appressorium formation in the Δ*Moptps* mutants.

To further evaluate the hypothesis that the decreased rate of hyphal growth in infected cells and reduced virulence of the Δ*Moptp2* mutant was due to a lack of ROS scavenger production by the host plant, an excised leaf-sheath assay was performed using DPI. After incubation at 28°C for 36 hr, cells were observed under a light microscope. Without DPI treatment, the infectious hyphae of Δ*Moptp2* mutants showed restricted growth in the primary infected cells, while those of Guy11 spread into the adjacent cells. Upon treatment with 0.5 μM DPI, the *Moptp2* infectious hyphae spread into neighboring cells. Similar results were observed by using the *Aspergillus niger* catalase (CAG, Sigma) (*Tanabe et al., 2009*) that hydrolyzes ROS (*Figure 6D, E and F*). This observation indicated that MoPtp2 has an effect on suppressing accumulation of host-derived ROS during infection.

## MoPtp1/2 are essential in suppressing MoOsm1 phosphorylation and function

To study whether the ROS accumulation caused by the deletion of *MoPTP1* and *MoPTP2* was dependent on their phosphatase activity, we generated the phosphatase activity inactivation mutant of MoPtp1 and MoPtp2 (*Figure 7—figure supplement 1*). The pathogenicity assay demonstrated that the phosphatase activity dead mutants showed a similar phenotype to the Δ*Moptp1* and Δ*Moptp2* mutants (*Figure 7A*). We further identified the phosphorylation pattern of MoOsm1 in these mutations. The results showed that, in both Δ*Moptp1/MoPTP1*$^{Δptpc}$ and Δ*Moptp2/ MoPTP2*$^{Δptpc}$ mutants, the band of MoOsm1-GFP migrated as the phosphorylated MoOsm1-GFP protein treated with PI and as in Δ*Moptp1/2* mutants (*Figure 7B*), indicating that the phosphatase activity is critical for the dephosphorylation of MoOsm1. We also detected the phosphorylation levels of MoOsm1 using the antiphospho-p38 antibody and observed increased MoOsm1 phosphorylation in both the Δ*Moptp1* and Δ*Moptp2* mutants (*Figure 7—figure supplement 2*). In addition, the phosphorylation of MoOsm1 remained at high even after $H_2O_2$ treatment at 60 min in the Δ*Moptp2* mutants, while the wild type dropped to a low level at 10 min (*Figure 7—figure supplement 2*). Then we observed the localization of MoOsm1 in the Δ*Moptp1*, Δ*Moptp2* and also Δ*Moptp1/MoPTP1*$^{Δptpc}$ and Δ*Moptp2/MoPTP2*$^{Δptpc}$ mutants. MoOsm1 was present in both the cytosol and the nucleus evenly, similar to that of the wild type. When treated with $H_2O_2$, MoOsm1 showed an enhanced nuclear translocation pattern. At 30 min following $H_2O_2$ treatment, MoOsm1 showed a nucleus to cytoplasm shifting in the wild-type strain but not in the Δ*Moptp*2 and Δ*Moptp2/MoPTP2*$^{Δptpc}$ mutants (*Figure 7C*). These results further supported that MoPtp1/2-mediated dephosphorylation of MoOsm1 controls its nuclear-cytoplasm translocation that is important for the oxidation stress response in *M. oryzae*. As MoPtp1/2 function on the dephosphorylation of MoOsm1, we observed the formation of homodimer in MoOsm1 in these mutants. We found that the dimer exists in either Δ*Moptp1* or Δ*Moptp2* mutants; however, in the Δ*Moptp1*Δ*Moptp2* double mutant, we observed monomerized MoOsm1 only, which is similar to that of MoOsm1$^{Y173D}$ (*Figure 7D*).

Additionally, the interaction between MoAtf1 and MoTup1 is also dependent on the phosphorylation of MoOsm1. When detected using co-IP, we found that the MoAtf1-MoTup1 complex disassociates in the Δ*Moptp2* and Δ*Moptp2/MoPTP2*$^{Δptpc}$ mutants; however, MoAtf1 remains interacting with MoTup1 in the Δ*Moptp1* mutant (*Figure 7E*). These results suggested that MoPtp1 and MoPtp2 are involved in suppressing the overactivation of MoOsm1 upon ROS stress and that MoPtp2 may play a major role in the dephosphorylation of MoOsm1 in *M. oryzae*.

## Identification of MoPtp1 and MoPtp2 as MoAtf1 targets

To identify MoAtf1 binding *cis*-elements in target promoters, we analyzed the ChIP data using the multiple EM for motif elicitation (MEME) program for 9 bp cis-elements containing the TGAY(C/T)R (G/A)W(T/A) motif (*Figure 8—figure supplement 1A*). We found that this *cis*-element exists in the promoters of MoAtf1-binding oxidoreduction-pathway genes (*Figure 8—figure supplement 1B*). According to the ChIP data, MoAtf1 binds with the promoter of *MoPTP2*, but not *MoPTP1* (*Figure 8—figure supplement 1C*). However, the same *cis*-element was found in the promoter of both *MoPTP1* and *MoPTP2* (*Figure 8—figure supplement 1D*). To verify the relationship between MoAtf1 and MoPtp1/2, we observed the expression of *MoPTP1* and *MoPTP2* at various developmental stages that showed the transcription levels of *MoPTP1* and *MoPTP2* were dramatically decreased in the Δ*Moatf1* mutant (*Figure 8A and B*). DNA containing the promoter sequence of

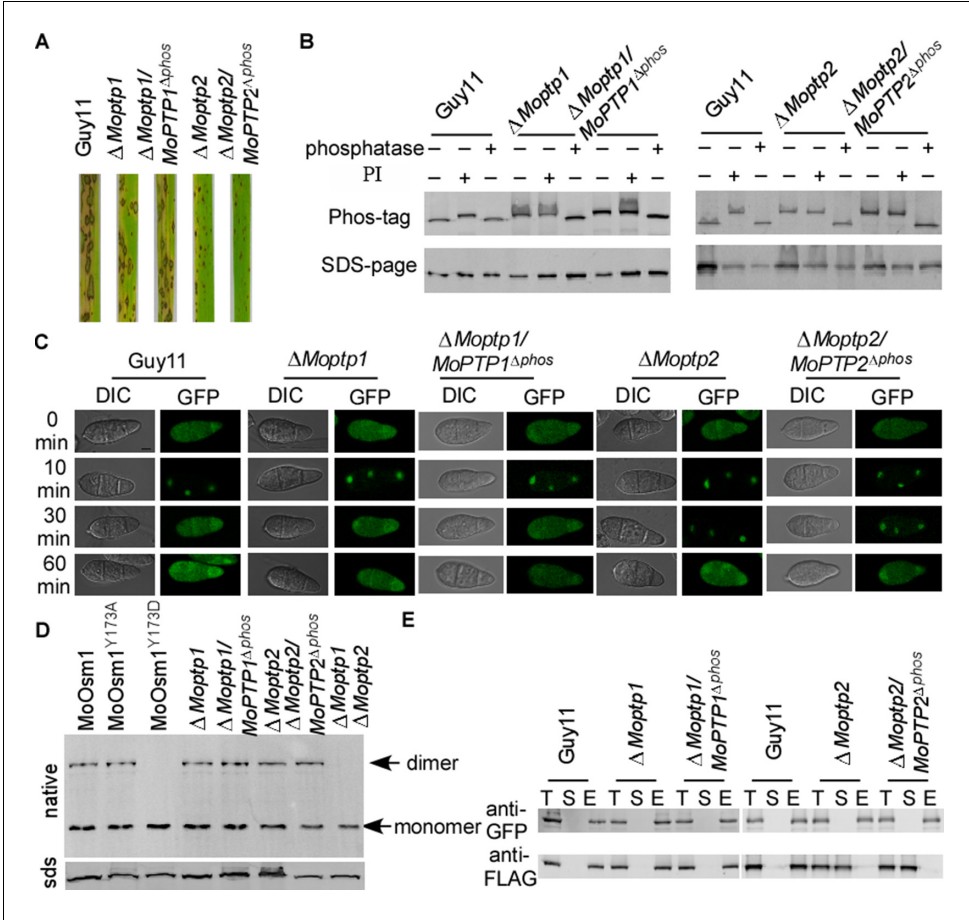

**Figure 7.** MoPtps-mediated dephosphorylation of MoOsm1 leads to its nuclear exporting and monomerization. (A) Pathogenicity assay. Conidial suspensions were sprayed onto two-week-old rice seedlings (CO-39). Diseased leaves were photographed after 7 days of inoculation. (B) Dephosphorylation of MoOsm1 dependent on the phosphatase activity of MoPtp1/2. MoOsm1-GFP proteins of various sources treated with alkaline phosphatase and phosphatase inhibitors were separated by $Mn^{2+}$-Phos-tag SDS-PAGE. (C) Fluorescence observation of conidia untreated (top panels) and treated with 5 mM $H_2O_2$ for 10, 30, and 60 min (lower panels). Bar = 5 μm. (D) The total proteins of MoOsm1-GFP/ΔMoosm1, MoOsm1$^{Y173A}$-GFP/ΔMoosm1, MoOsm1$^{Y173D}$-GFP/ΔMoosm1, ΔMoptp1, ΔMoptp1/MoPTP1$^{Δptpc}$, ΔMoptp2, and ΔMoptp2/MoPTP2$^{Δptpc}$ were subjected to Native-PAGE followed by immunoblotting analysis. (E) Western blot analysis of MoAtf1-MoTup1 interactions in ΔMoptp1/2 mutant strains. The presence of MoAtf1-GFP and MoTup1-FLAG was detected with the anti-GFP and anti-FLAG antibodies, respectively. T: total proteins; S: suspension proteins; and E: elution proteins.

The online version of this article includes the following figure supplement(s) for figure 7:

**Figure supplement 1.** A comparison of MoPtp1 and MoPtp2 functional domains.
**Figure supplement 2.** MoOsm1 phosphorylation levels in ΔMoptp1/2 mutants under oxidative stress.

*MoPTP1* and *MoPTP2* was retarded by the addition of the purified MoAtf1 protein and this retardation increased drastically as the amount of MoAtf1 increased (*Figure 8C and D*). To further confirm this binding, we used unlabeled DNA to compete with the Alex660-labeled DNA for the purified MoAtf1 protein's binding sites and found that the binding of labeled DNA with MoAtf1 was decreased significantly with the rise in unlabeled DNA (*Figure 8E and F*). Further, we generated the putative cis-elements deletion promoter of *MoPTP1* and *MoPTP2* to confirm the binding with MoAtf1. After separated by polypropylene gel, we found that the band of the digoxin (DIG)-labeling oligomer binding with *MoPTP1* and *MoPTP2* promoters migrated slower than that of free DNA, and the mobility decreased with increasing concentration of MoAtf1 (*Figure 8G and H*). When deleted the putative cis-elements from *MoPTP1* and *MoPTP2* promoters, the oligomer band migrated as fast

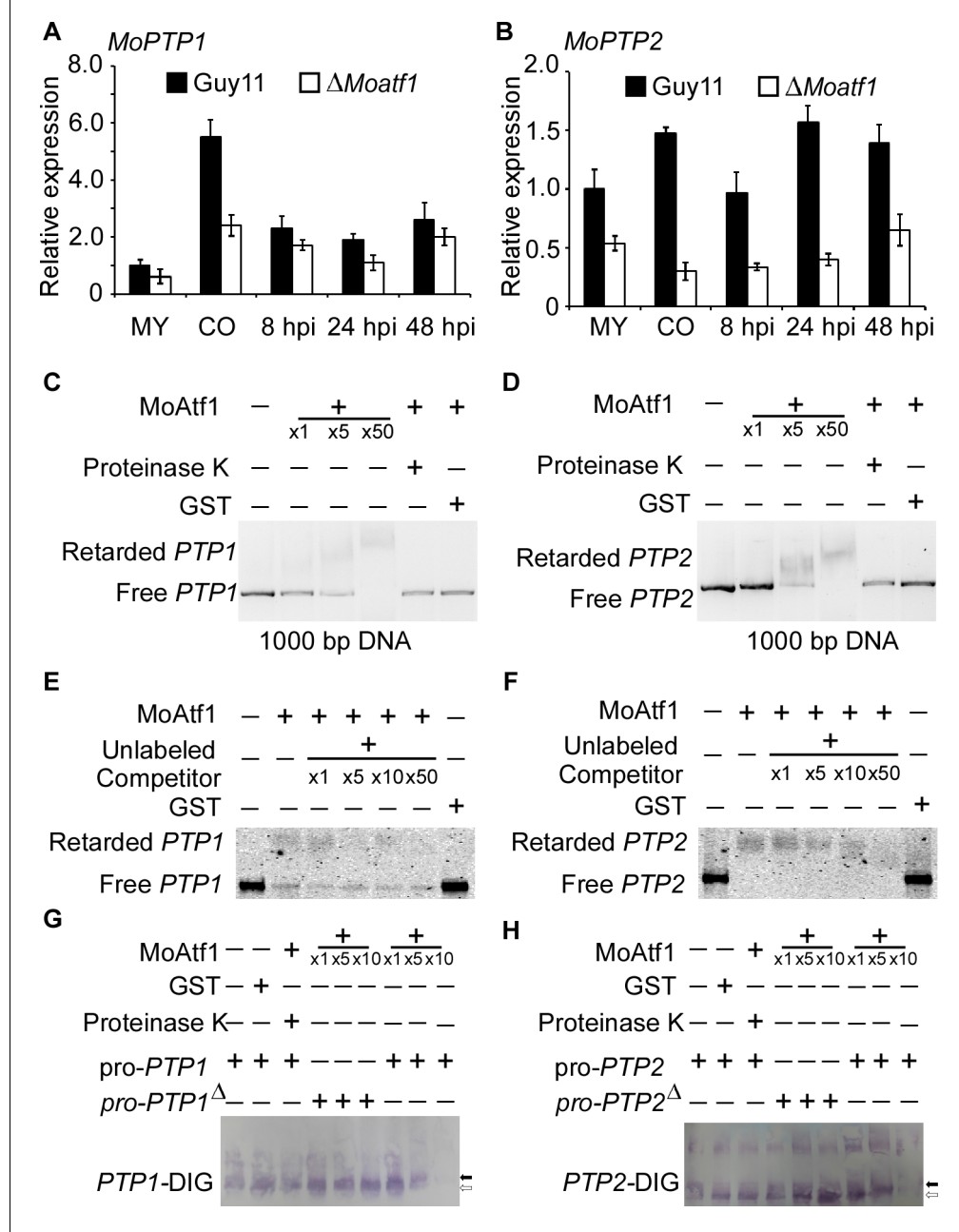

**Figure 8.** MoAtf1 binds to the promoter regions of *MoPTP1* and *MoPTP2*. (**A**) Expression analysis of *MoPTP1* in the Δ*Moatf1* mutant and wild-type strains. The expression of *MoPTP1* was analyzed by qRT-PCR and normalized to that of actin gene (MGG_03982). MY represents mycelium and CO represents conidium. Error bars represent the standard deviations and asterisks represent significant differences (p<0.01). (**B**) Expression analysis of *MoPTP2* in the Δ*Moatf1* mutant and wild-type strains. (**C**) and (**D**) The full-length promoter sequence of *MoPTP1* and *MoPTP2* was incubated in the absence (leftmost lane) or presence (second to the fourth lane with increasing amounts of MoAtf1) of purified MoAtf1 and GST protein (rightmost lane). The proteinase K was added after the incubation of MoAtf1 with the DNA (fifth lane). DNA-protein complexes were separated by electrophoresis on a 1.2% agarose gel. (**E**) and (**F**) The Alex660-labeled full-length DNA of promoter was incubated in the absence (leftmost lane) or presence (second to the sixth lane) of the purified MoAtf1 and GST protein (rightmost lane). Unlabeled DNA was added as a binding competitor with increasing amounts in lanes from third to sixth. DNA-protein complexes were separated by electrophoresis on a 1.2% agarose gel. (**G**) The DIG-labeled promoter of *MoPTP1* was incubated with an increasing amount of MoAtf1 prior to separation by PAGE. pro-*PTP1*: full-length promoter. pro-*PTP1*Δ: the cis-element deletion promoter. The white arrow represents free DNA, the black arrow represents migrated DNA.
*Figure 8 continued on next page*

*Figure 8 continued*

(H) The DIG-labeled promoter of *MoPTP2* was incubated with an increasing amount of MoAtf1. The white arrow represents free DNA, the black arrow represents migrated DNA.

The online version of this article includes the following figure supplement(s) for figure 8:

**Figure supplement 1.** The ChIP assay identifies *MoPTP1* and *MoPTP2* as putative targets of MoAtf1.

as the free DNA (*Figure 8G and H*). These results indicated that MoAtf1 regulates the expression of *MoPTP1* and *MoPTP2* by directly targeting their promoters.

## Phosphorylation of MoAtf1 is important for transcription initiation under oxidation stress

Given ROS burst activates MoOsm1, we hypothesized that MoPtp1/2-mediated dephosphorylation was also a response to oxidation stress. We then next examined the expression of *MoPTP1* and *MoPTP2* genes and found that the expression was induced upon $H_2O_2$ exposure. As MoAtf1 binds to the promoters of *MoPTP1* and *MoPTP2*, we further analyzed whether the phosphorylation of MoAtf1 affects this expression. Even if the transcription levels were low, the expression of *MoPTP1* and *MoPTP2* genes increased and peaked after being treated with $H_2O_2$ for 10 min in the $\Delta Moosm1$/*MoATF1*$^{S124D}$ strain, similar to the wild-type strain (*Figure 9A and B*). In the $\Delta Moosm1$ mutants, whose phosphorylation level of MoAtf1 was low, however, the expression levels of *MoPTP1* and *MoPTP2* genes remain unchanged (*Figure 9A and B*). In addition, we evaluated the protein levels of MoPtp1 and MoPtp2 under oxidation stress. After treated with $H_2O_2$ for 10 min, the amount of MoPtp1 and MoPtp2 showed no significant difference in the $\Delta Moosm1$ mutant (*Figure 9C and D*), but the protein levels of MoPtp1 and MoPtp2 were induced significantly in the $\Delta Moosm1$/*MoATF1*$^{S124D}$ strain under stress (*Figure 9E and F*). Taken together, these results indicated that the phosphorylation of MoAtf1-mediated by MoOsm1 is required for the expression of *MoPTP1* and *MoPTP2* in response to oxidation stress.

## Discussion

Upon pathogen infection, plants rapidly activate PTI through ROS burst and callose deposition. Meanwhile, pathogens secrete numerous extracellular proteins into the plant to circumvent host immunity (*Doehlemann and Hemetsberger, 2013*). Previous studies have identified several effector proteins, including those involved in oxidation-related functions; however, how these factors initiate their function to counter host immunity remains not fully understood. In this study, we discovered that MoOsm1 phosphorylates MoAtf1 that in turn dissociates from MoTup1 to regulate oxidation stress response pathways. We further revealed that phosphorylated MoAtf1 promotes the expression of MoPtp1/2 that could dephosphorylate MoOsm1, completing a feedback loop that controls the virulence of *M. oryzae*. Our results demonstrated MoOsm1-MoPtps-mediated feedback loop represents a previously unknown mechanism that balances the response to ROS stress and the hemibiotrophic growth of *M. oryzae*.

For the pathogen, ROS seems like a double-edged sword. During appressorial penetration, *M. oryzae* accumulates high levels of endogenous ROS to strengthen the appressorium cell wall. (*Egan et al., 2007*). Here, ROS accumulation is regulated by two fungal NADPH oxidases, which themselves are important for appressorium-mediated cuticle penetration (*Dagdas et al., 2012*). However, ROS is also one of the earliest responses by plants to microbial colonization and function a potent defense mechanism that limits fungal biotrophic growth (*Torres, 2010*; *Torres et al., 2006*). Rboh, which encodes NADPH oxidase, plays a key role in generating ROS upon pathogen challenge (*Wong et al., 2007*). Among them, both OsrbohA and OsrbohB are important for the production of ROS when rice are subjected to stresses (*Wong et al., 2007*). In addition, during *M.oryzae* infection, *OsRBOHA* and *OsRBOHB* were significantly induced in WT plants at 24 hpi, but decreased afterward (*Yang et al., 2017*). As the oxidative burst reaction occurs rapidly, how can the pathogen quickly respond to this stress? In *M. oryzae*, the MoOsm1-mediated osmoregulation pathway is essential for responding to oxidative stress, in addition to the previously demonstrated hyperosmotic stress. The deletion of *MoOSM1* was shown to result in high sensitivity to oxidative stress (*Dixon et al., 1999*).

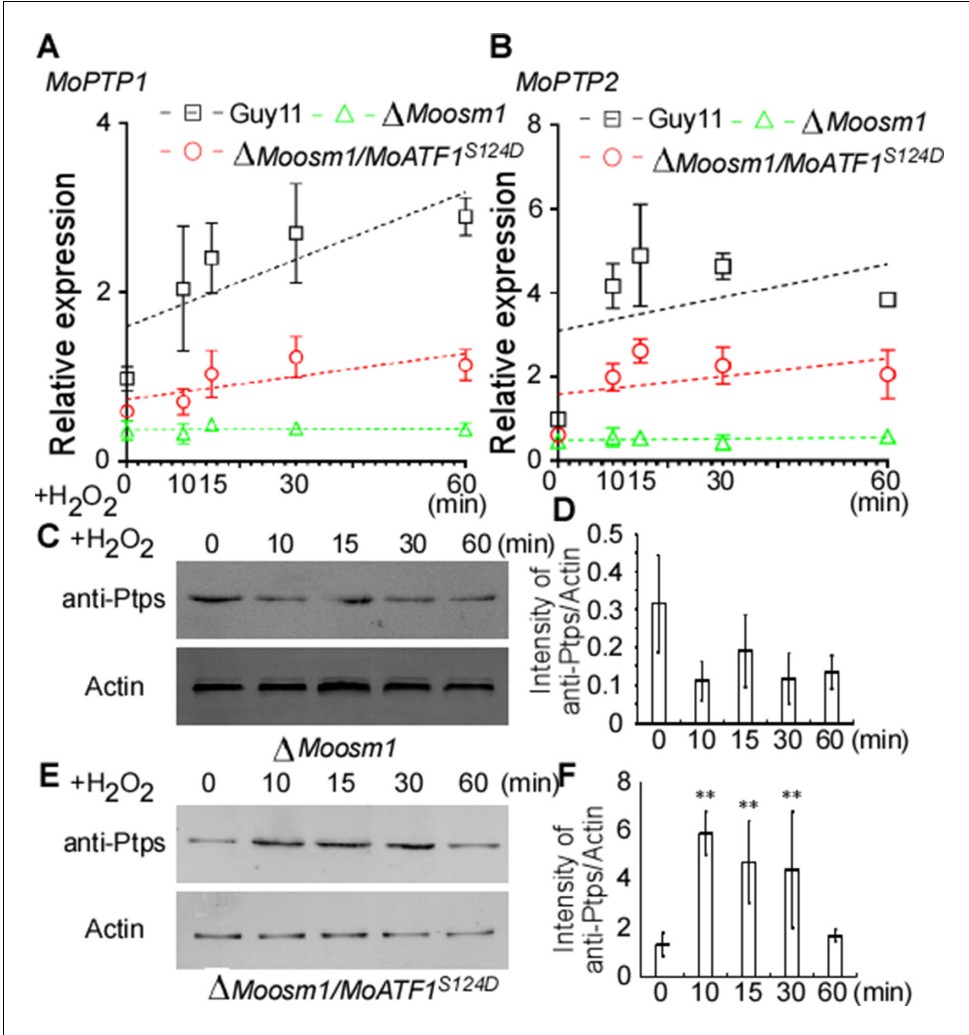

**Figure 9.** MoAtf1 phosphorylation controls the transcription of *MoPTP1* and *MoPTP2* in response to oxidative stress. (**A**) and (**B**) *MoPTP1* and *MoPTP2* expression analysis in Guy11, Δ*Moosm1* mutant, and Δ*Moosm1/MoAtf1^{S124D}* strains treated with $H_2O_2$ for 15, 30, and 60 min. Three independent biological experiments were performed, with three replicates each time, and yielded similar results in each independent biological experiment. Dotted lines represent the expression of *MoPTP1* and *MoPTP2* in these strains under $H_2O_2$ stress. Error bars represent standard deviation. (**C**) Total proteins were extracted from Δ*Moosm1* treated with $H_2O_2$ for 15, 30, and 60 min. MoPtp1/2 was detected by western blotting analysis using anti-ptp1/2 antibodies. An anti-Actin antibody was used as control. (**D**) Western blotting bands were quantified with an ODYSSEY infrared imaging system (application software Version 2.1). Bars denote standard errors from three independent experiments. Asterisks indicate significant differences (Duncan's new multiple range test p<0.01). (**E**) Total proteins were extracted from Δ*Moosm1/MoAtf1^{S124D}* strain for MoPtp1/2 detection. (**F**) Western blotting bands of MoPtp1 in Δ*Moosm1/MoAtf1^{S124D}* strain was quantified with an ODYSSEY infrared imaging system.

We here showed that MoOsm1 phosphorylation also increases during the early stage of infection. Given that ROS burst is one of the earliest PTI responses during infection, ROS burst may activate osmoregulation MAPK signaling through MoOsm1 phosphorylation. Intriguingly, oxidative stress was also found to result in MoOsm1 cytoplasm to nucleus translocation.

How MoOsm1 translocated into the nucleus in response to oxidative stress? Previous studies showed that the dephosphorylation of MoHat1 led to its interaction with MoSsb1, causing a nucleus to cytoplasm translocation (*Yin et al., 2019*). Studies also showed that MoAp1 accumulated in the nucleus under oxidative stress (*Guo et al., 2011*). In *S. cerevisiae*, Ap1 protein forms disulfide bonds that dampen the recognition of the nuclear export protein Crm1, leading to its accumulation in the

nucleus (*Kuge et al., 2001*) which prompted us to suggest a relevance between MoAtf1 phosphory-lation and its nuclear localization. *Arabidopsis* regulatory protein NPR1 is involved in salicylic acid (SA)-mediated defense response in which SA triggers cytoplasmic NPR1 oligomer release that is required for its nuclear import (*Mou et al., 2003*). The SNF1-related protein kinase SnRK2.8 phosphorylate NPR1 prior to its nuclear trans-localization (*Lee et al., 2015*). In addition, the nitric oxide-induced S-nitrosoglutathione also causes NPR1 nuclear translocation (*Lindermayr et al., 2010*). Given these findings, we hypothesized and demonstrated that (1) phosphorylation of MoOsm1 inhibits the recognition of the nuclear export proteins causing its retention in the nucleus; (2) MoOsm1 forms homodimers in the cytoplasm, but MoOsm1 oligomers are released and imported into the nucleus when MoOsm1 was phosphorylated. We also showed that MoOsm1 forms homodimers that disintegrate upon phosphorylation of Y173 (*Figure 3D–F*). Further combined with the results that protein dimerization occurs only in the cytoplasm (*Figure 3G*) and MoOsm1 phosphorylation on Y173 leads to monomerization and nuclear localization, we thus considered that phosphorylation happening prior to nuclear import instead of the misrecognition of phosphorylated MoOsm1 in the nucleus that caused the nuclear accumulation of MoOsm1. Collectively, we concluded that the monomerization of MoOsm1 is important for its nuclear accumulation and Y173 phosphorylation is important for monomerization. Given the evidence indicating that the nuclear localization of MoOsm1 was more stable in the ΔMoptps mutant than Guy11 under ROS stress, it is plausibly that MoPtps function in the recycling of MoOsm1 to the cytoplasm. MoOsm1 recycling may have two benefits: (1) shutdown of phosphorylated MoOsm1 mediates signaling pathways that switch off virulence attack. (2) recycled MoOsm1 in the cytoplasm might respond quickly to external stress as there is no need for protein re-synthesing.

MoOsm1 co-localizes with and activates the transcription factor MoAtf1 through protein phosphorylation in the nucleus. MoAtf1 is one of the bZIP transcription factors control gene expression during plant infection (*Kim et al., 2009*; *Proft et al., 2001*; *Tang et al., 2015*). Here, we found that MoAtf1 positively regulates genes involved in oxidation response pathways (*Figure 5—figure supplement 3*). During infection, the expression of the oxidation responsive genes was significantly upregulated (*Figure 5*). This led us to identify MoTup1 as one of the proteins interacting with MoAtf1. MoTup1 was recently characterized in *M. oryzae*, and its deletion caused decreased pathogenicity (*Chen et al., 2015*). This study, together with those of other model organisms (*Chen et al., 2013*; *García-Sánchez et al., 2005*; *Malavé and Dent, 2006*; *Smith and Johnson, 2000*), suggests that Tup1 proteins do not bind directly to DNA but rather are brought to the promoters *via* interactions with sequence-specific regulatory proteins. We speculate that MoAtf1 recruits MoTup1 to repress transcription. Once MoAtf1 phosphorylates and no longer binds with MoTup1, MoAtf1 upregulates or initiates the transcription of target genes, including those involved in oxidation regulated pathways.

During the interaction between *M. oryzae* and the host, the host induces ROS burst during the biotrophic growth stage of the fungus. Under ROS stress, the pathogen activates MoOsm1-mediated pathways that in turn activates the transcription factor MoAtf1. The phosphorylation of MoAtf1 causes the disintegration of the MoAtf1-MoTup1 complex to induce the expression of oxidation regulation pathway genes to further enhance virulence of *M. oryzae*. Phosphorylated MoAtf1 also initiates the transcription of the *MoPTP1/2* genes to further dephosphorylate MoOsm1. Once the ROS stress was circumvented, such as that at the necrotrophic growth stage, MoPtps- mediated dephosphorylation of MoOsm1 shut off phosphor-regulatory circuitry to control the virulence (*Figure 10*).

## Materials and methods

### Strains and culture conditions

The *M. oryzae* Guy11 strain was used as a wild type (WT) in this study. All strains were cultured on complete medium (CM) for 3–15 days in the dark at 28℃ (*Liu et al., 2016*; *Talbot et al., 1993*). Mycelia were harvested from the liquid CM media with or without additional treatment for DNA, RNA, and total protein extractions. For conidia production, strains were maintained on straw decoction and corn (SDC) (100 g of straw, 40 g of corn powder, 15 g of agarin 1 l of distilled water) agar media at 28℃ for 7 days in the dark followed by 3 days of continuous illumination under fluorescent light (*Qi et al., 2016*).

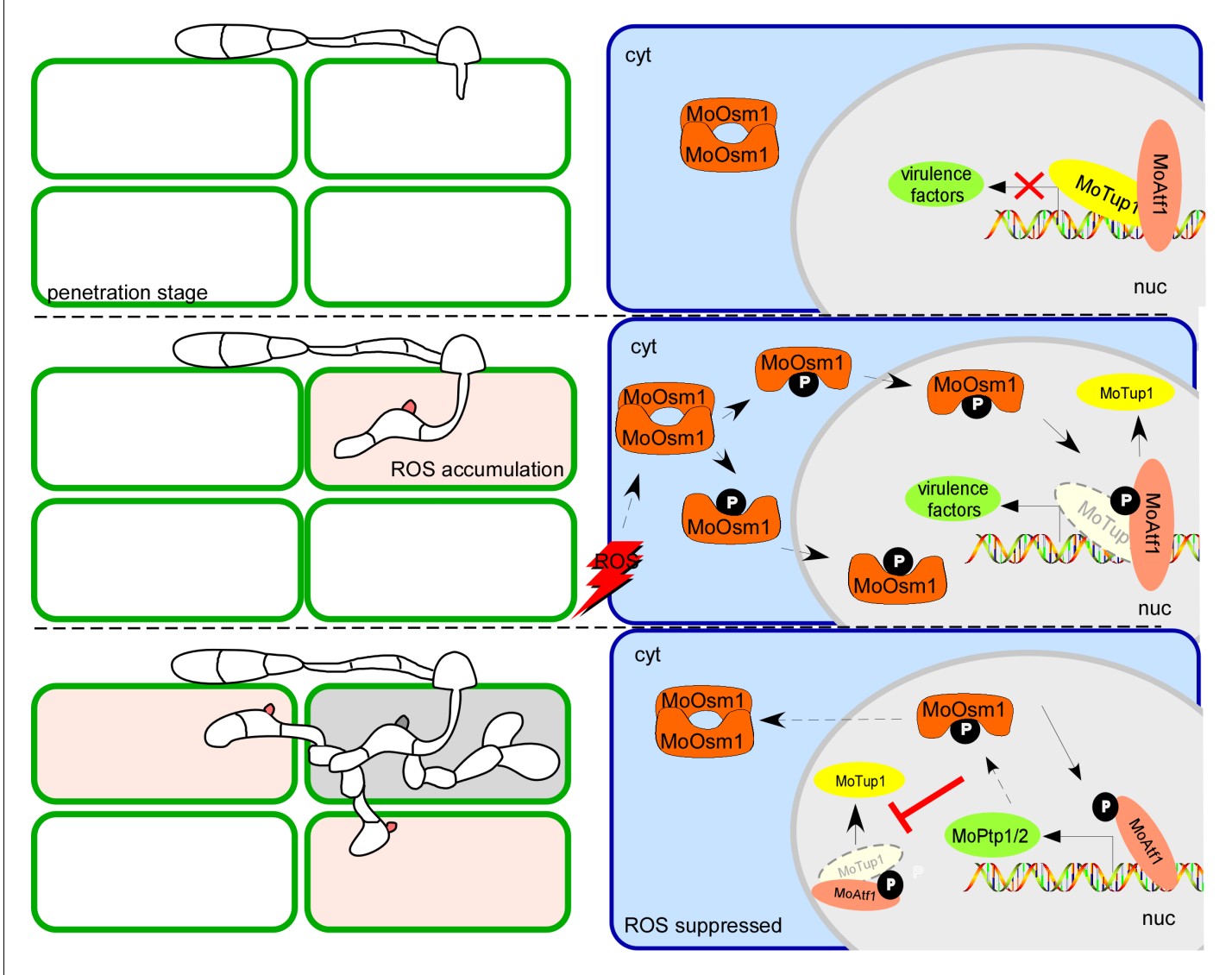

**Figure 10.** A proposed model depicting MoOsm1/MoAtf1/MoTup1/MoPtp1/2 mediated ROS signaling and responses to host immunity. Rice generates immunity, including ROS burst during its interaction with *M. oryzae*. Once host perception, *M. oryzae* induces MoOsm1 phosphorylation that disintegrates MoOsm1 dimerization leading to enhanced nuclear translocation of MoOsm1. MoOsm1 phosphorylates MoAtf1 uncoupling MoAtf1-MoTup1 interaction that induces the expression of oxidation regulation pathway genes. At the same time, the phosphorylated MoAtf1 promotes the expression of two phosphatases, *MoPTP1* and *MoPTP2*, that dephosphorylate MoOsm1 to suppress MoAtf1-MoTup1 dissociation. MoPtp1/2-mediated MoOsm1 dephosphorylation provides an act balancing infection and its hemibiotrophic growth in rice.

## Targeted gene deletion and transformation

The *MoPTPs* gene deletion mutants were generated using the standard one-step gene replacement strategy. First, two fragments with 1.0 kb of sequences flanking the targeted gene were PCR amplified with primer pairs. The resulting PCR products were digested with restriction endonucleases and ligated with the hygromycin-resistance cassette (*HPH*) released from pCX62. Finally, the recombinant insert was sequenced. The 3.4 kb fragment, which includes the flanking sequences and the HPH cassette, was amplified and transformed into Guy11 protoplasts. Putative mutants were first screened by PCR and later confirmed by Southern blotting analysis (*Figure 6—figure supplement 1*). Fragments for mutant complementation were amplified by PCR and inserted into pYF11 or pHZ126 before being introduced into the mutant strains through PEG-mediated transformation.

## Virulence assay

Conidia harvested from 10-day-old SDC agar cultures were filtered through two layers of Miracloth and resuspended to a concentration of $5 \times 10^4$ spores/ml in a 0.2% gelatin solution. Two-week-old seedlings of rice (*Oryza sativa* cv. CO39 and K23) was used for pathogenicity assays. For spray inoculation, 5 ml of a conidial suspension of each treatment was sprayed onto rice with a sprayer. Quantification of lesion types (0, no lesion; 1, pinhead-sized brown specks; 2, 1.5 mm brown spots; 3, 2–3 mm gray spots with brown margins; 4, many elliptical gray spots longer than 3 mm; 5, coalesced lesions infecting 50% or more of the leaf area) were measured according to *Wang et al., 2013* Conidial germination and appressorium formation were measured on a hydrophobic surface as previously described (*Qi et al., 2012*). Appressorium induction and formation rates were also obtained as described previously (*Li et al., 2017a*).

For infection, conidia were harvested from 10-day-old SDC agar cultures, filtered, and resuspended to a concentration of $5 \times 10^4$ spores/ml in a 0.2% (w/v) gelatin solution. For the leaf assay, leaves from two-week-old seedlings of rice (*Oryza sativa* cv. CO39 or K23) and 7-day-old seedlings of barley were used for spray inoculation. For rice leaves, 5 ml of a conidial suspension of each treatment was sprayed. Inoculated plants were kept in a growth chamber at 25°C with 90% humidity and in the dark for the first 24 hr, followed by a 12/12 hr light/dark cycle. Lesion formation was observed daily and recorded by photography 7 days after inoculation (*Yin et al., 2020*).

For DAB staining, the leaf sheaths were immersed in 1 mg/ml solution of DAB in a buffer (pH = 3.8) at the indicated time after inoculation with *M. oryzae*. Samples were incubated at room temperature for 8 hr in the dark. When the brown spots appeared clearly, samples were bleached ethanol: acetic acid (95:5) for 1 hr. Images were captured using a microscope (Zeiss, Axio Observer A1).

## Quantitative RT-PCR analysis

For qRT-PCR, total RNA was reverse transcribed into first-strand cDNA using the oligo (dT) primer and HiScript II Q select RT SuperMix for qPCR (Vazyme, Nanjing, R233-01). The qRT-PCR was run on the Applied Biosystems 7500 Real-Time PCR System with ChamQ SYBR qPCR Master Mix (Vazyme, Nanjing, Q311-02). Normalization and comparison of mean Ct values were performed as previously described (*Yin et al., 2020*).

## Epifluorescence microscopy

*M. oryzae* cells (conidia) expressing fluorescent protein-fused chimera were incubated under appropriate conditions. The constructs including MoOsm1-GFP, MoAtf1-GFP, and other phosphorylation mutations were transformed into Δ*Moosm1* mutant, Δ*Moatf1* mutant or the wild-type Guy11 strain. Epifluorescence microscopy was performed using a Zeiss LSM710 (63x oil) microscope. H1-RFP was introduced into the MoOsm1-GFP transformants to visualize the nucleus.

## Yeast two-hybrid (Y2H) and bimolecular fluorescence complementation assays

cDNA was respectively amplified with Super Fidelity DNA Polymerase. Amplified products were cloned into pGBKT7 or pGADT7 vectors, respectively. After sequence verification, they were introduced into yeast AH109 strain. Transformants grown on synthetic medium lacking leucine and tryptophan (SD–Leu–Trp) were transferred to synthetic medium lacking leucine, tryptophan, and histidine (SD–Leu–Trp–His).

For the BiFC assay, the cYFP-*MoOSM1* fusion construct was generated by cloning *MoOSM1* into pHZ68. Similarly, *MoOSM1-nYFP*, *MoOSM1^{Y173D}-nYFP*, *MoOSM1^{Y173A}-nYFP* and *MoATF1-nYFP* fusion constructs were generated into pHZ65, respectively. Construct pairs were introduced into the protoplasts of Guy11, respectively. Transformants resistant to both hygromycin and zeocin were isolated and confirmed by PCR.

## Co-immunoprecipitation (co-IP) assay

To confirm the interactions among MoOsm1-MoOsm1, MoOsm1-MoAtf1, and MoAtf1-MoTup1 in vivo, MoOsm1-GFP, MoOsm1-FLAG, MoTup1-GFP, MoAtf1-GFP, and all of the mutation protein fusion constructs were prepared by the yeast homologous recombination transformation. Different

pairs of specific constructs were co-transformed into the protoplasts of the WT strain. Total hyphae proteins were isolated from different positive transformants and incubated with anti GFP agarose (Chromo Tek, gta-20) at 4°C for 2 to 12 hr with gently shaking. Proteins bound to the beads were eluted after a series of washing steps by 1 × PBS. Elution buffer (200 mM glycine, pH 2.5) and neutralization buffer (1 M Tris, pH 10.4) were used for the elution process. Total, suspension, and eluted protein were analyzed by western blot using GFP (mouse, 1:5000; Abmart, 293967) or FLAG (mouse, 1:5000; Abmart, M20018) specific antibodies.

## GST-pull down

GST, GST-MoAtf1, GST-MoAtf1$^{S124D}$, and His-MoTup1 were expressed in *Escherichia coli* BL21-CodonPlus (DE3) cells. Cells were lysed in lysis buffer (50 mM Tris, pH 8.0, 50 mM NaCl, 1 mM PMSF [Beyotime Biotechnology, ST506-2]) with a sonicator (Branson). Samples were centrifuged (13,000 g, 10 min) and the supernatants were transferred to a new 1.5 ml tube and stored at −70°C. The GST, GST-MoAtf1, and GST-MoAtf1$^{S124D}$ supernatants were then mixed with 30 μl glutathione sepharose beads (GE Healthcare, 10265165) and incubated at 4°C for 2 hr. The recombinant GST, GST-MoAtf1 or GST-MoAtf1$^{S124D}$-bound to glutathione sepharose beads were incubated with *E. coli* cell lysate containing His-MoTup1 at 4°C for another 4 hr. Finally, the beads were washed with buffer (50 mM Tris, pH 8.0, 50 mM NaCl, 1 mM PMSF, 1% Triton X-100) five times and eluted from the beads. Eluted proteins were then analyzed by immunoblot (IB) with monoclonal anti-His and monoclonal anti-GST antibodies (*Li et al., 2017a*), respectively.

## Protein extraction and western blot analysis

Mycelia were ground into a fine powder in liquid nitrogen for total protein extraction and resuspended in 1 ml lysis buffer (10 mM Tris-HCl, pH7.5, 150 mM NaCl, 0.5 mM EDTA, 0.5% NP-40) with 2 mM PMSF and proteinase inhibitor cocktail. The lysates were placed on the ice for 30 min and shaken once every 10 min. Cell debris was removed by centrifugation at 13, 000 g for 10 min at 4°C. The lysates were collected as to total proteins (*Liu et al., 2019*). For GFP- tagged protein detection, samples were separated by 8% SDS-PAGE and followed by western blotting analysis. For detecting phosphorylated MoOsm1, a p38 MAP kinase orthologue, the anti-pp38 (CST:9215S) and anti-p38 (CST:9212S) antibodies were used. For detecting MoPtp1 and MoPtp2, the anti-ptp1B antibody (ab244207) was used. Blot signals were detected and analyzed using the ODYSSEY infrared imaging system (Version 2.1).

## Phosphorylation analysis through Phos-tag gel electrophoresis

The MoAtf1-GFP fusion construct was introduced into the wild type (Guy11) and Δ*Moosm1*, and MoOsm1-GFP was introduced into Guy11, Δ*Moptp1*, and Δ*Moptp2* mutant strains, respectively. The total protein extracted from mycelium was resolved on 8% SDS-polyacrylamide gels prepared with 50 μM acrylamide-dependent Phos-tag ligand and 100 μM MnCl$_2$ as described (*Li et al., 2017a*). Gel electrophoresis was performed with a constant voltage of 80 V for 3–6 hr. Before transferring, gels were equilibrated in transfer buffer with 5 mM EDTA for 20 min two times and followed by transfer buffer without EDTA for another 20 min. Protein transfer from the Mn$^{2+}$-phos-tag acrylamide gel to the PVDF membrane was performed for ~36 hr at 80 V at 4°C, and then the membrane was analyzed by western blotting using the anti-GFP antibody (*Li et al., 2017b*).

## In vitro phosphorylation analysis

The GST-MoAtf1, GST-MoAtf1$^{S124A}$, and His-MoOsm1 were expressed in *E. coli* BL21-CodonPlus (DE3) cells and purified as described in the GST-pull down assays. The rapid and cost-effective fluorescence detection in tube (FDIT) method was used to analyze protein phosphorylation in vitro (*Jin and Gou, 2016*). The Pro-Q Diamond Phosphorylation Gel Stain, known as a widely used phosphor-protein gel-staining fluorescence dye, was used in this assay. For protein kinase reaction, 2 μg GST-MoAtf1, GST-MoAtf1$^{S124A}$ was mixed with MoOsm1 in a kinase reaction buffer (100 mM PBS, pH 7.5, 10 mM MgCl$_2$, 1 mM ascorbic acid), with the appearance of 50 μM ATP at room temperature (RT) for 60 min, 10 folds of cold acetone was added to stop the reaction. For protein in tube staining, casein (Sango Biotech, T510256) was homogenized and suspended in Mili-Q water at the concentration of 0.2 μg/μl. For staining duration time analysis, 10 μl of casein was mixed with 100 μl

of Pro-Q Diamond (Thermo Fisher Scientific, P33301) and kept in the dark at RT for 1 hr. The protein was then precipitated with 10 volumes of cold acetone, kept in a −20°C freezer, and centrifuged at 13,200 g for 1 hr at 4°C. The supernatant was carefully drained out and discarded without touching the protein pellet. The pellet was rinsed with 0.5 ml of cold acetone and centrifuge to remove the supernatant twice. The pellet was air-dried and dissolved in 200 µl of Mili-Q water and moved to a black 96 well plate (Corning, 3925). Fluorescence signal at 590 nm (excited at 530 nm) was measured in a Cytation3 microplate reader (Biotek, Winooski, VT, USA) (*Yin et al., 2019*; *Yin et al., 2020*).

## Chromatin immunoprecipitation (ChIP)-Seq analyses

ChIP was performed according to the described protocol with additional modifications. Briefly, fresh mycelia were cross-linked with 1% formaldehyde for 15 min and then stopped with 125 mM glycine. The cultures were ground with liquid nitrogen and resuspended in the lysis buffer (250 mM, HEPES pH 7.5, 150 mM NaCl, 1 mM EDTA, 1% Triton, 0.1% Deoxy Cholate, 10 mM DTT) and protease inhibitor (Sangon Co., Shanghai, China, A100754). The DNA was sheared into ~500 bp fragments with 20 pulses of 10 s and 20 s of resting at 35% amplitude (Qsonica*sonicator, Q125, Branson, USA). After centrifugation, the supernatant was diluted with 10 × ChIP dilution buffer (1.1% Triton X-100, 1.2 mM EDTA, 16.7 mM Tris–HCl, pH 8.0 and 167 mM NaCl). Immunoprecipitation was performed using the monoclonal anti-GFP ab290 (Abcam, Cambridge, UK; 1:500 dilution) antibody. Following low salt wash (150 mM NaCl, 20 mM Tris-HCl (PH 8.0), 0.2% SDS, 0.5% TritonX-100, 2 mM EDTA), high salt wash (500 mM NaCl, 20 mM Tris-HCl (PH 8.0), 0.2% SDS, 0.5% TritonX-100, 2 mM EDTA), and LiCl and TE wash, DNA was eluted with elution buffer (1% SDS, 0.1 M NaHCO$_3$). The eluants were precipitated by ethanol after washing and digested with proteinase K, and sequenced on an Illumina HiSeq2500 (Genergy Bio, Shanghai, China). The accession number for RNA-seq data reported in this paper is GSE144389 in GEO datasets (http://www.ncbi.nlm.nih.gov/gds).

## Mass spectrometric analysis

Proteins were separated by 10% SDS-PAGE To identify phosphorylation sites. The gel bands corresponding to the targeted protein were excised, reduced with 10 mM of DTT and alkylated with 55 mM iodoacetamide. In-gel digestion (or elution digestion) was carried out with the trypsin (Promega, V5113), GLU-C (Wako, 050–05941), or chymotrypsin (Sigma-Aldrich, C6423) in 50 mM ammonium bicarbonate at 37°C overnight. The peptides were extracted using ultrasonic processing with 50% acetonitrile aqueous solution for 5 min and with 100% acetonitrile for 5 min. The extractions were reduced in the volume by centrifugation. A liquid chromatography-mass spectrometry (LC–MS) system consisting of a Dionex Ultimate 3000 nano-LC system (nano UHPLC, Sunnyvale, CA, USA), connected to a linear quadrupole ion trap Orbitrap (LTQ Orbitrap XL) mass spectrometer (ThermoElectron, Bremen, Germany) and equipped with a nanoelectrospray ion source, was used for our analysis. For LC separation, an Acclaim PepMap 100 column (C18.3 µm, 100 Å) (Dionex, Sunnyvale, CA, USA) capillary with a 15 cm bed length was used with a flow rate of 300 nL/min. Two solvents, A (0.1% formic acid) and B (aqueous 90% acetonitrile in 0.1% formic acid), were used to elute the peptides from the nanocolumn. The gradient ranged from 5% to 40% B in 80 min and from 40% to 95% B in 5 min, with a total run time of 120 min. The mass spectrometer was operated in the data-dependent mode to automatically switch between Orbitrap-MS and LTQ-MS/MS acquisition. Survey full-scan MS spectra (from m/z 350 to 1800) were acquired in the Orbitrap with a resolution r = 60,000 at m/z 400, allowing the sequential isolation of the top ten ions, depending on signal intensity. The linear ion trap fragmentation used collision-induced dissociation at a collision energy of 35 V. Protein identification and database construction were processed using Proteome Discoverer software (1.2 version, Thermo Fisher Scientific, Waltham, MA, USA) with the SEQUEST model.

## Electrophoretic mobility shift (EMSA) assays

The GST-MoAtf1 protein was expressed and purified from *E. coli* strain BL21 using the pGEX4T-2 construct. The DNA fragments from the promoter of *MoPTP1/2* were end-labeled with Alex660 by PCR amplification using the 5' Alex660-labeled primer. The purified protein was mixed with Alex660-labeled DNA, incubated for 20 min at 25°C in binding buffer, and separated by agarose gel electrophoresis. Gels were directly visualized using an LI-COR Odyssey scanner with excitation at 700 nm (*Wang et al., 2017*).

## Statistical analyses

Results were presented as the mean ± standard deviation (SD) of three biology repeats. The significant differences between samples were statistically evaluated by using SDs in SPSS 2.0. The significant differences between treatments with a single factor random grouping model were statistically determined by one-way analysis of variance (ANOVA) comparison and followed by the F-test if the ANOVA result is significant at p<0.01.

## Acknowledgements

This research was supported by the program of Natural Science Foundation of China (Grant No: 31972979, LX), NSFC-DFG (Grant No: 31861133017, ZZ), Natural Science Foundation of China (Grant No: and 31671979, ZX), and Innovation Team Program for NSFC (2017). We are grateful for Prof. Kunming Chen (North West Agriculture and Forestry University) and Prof. Mo Wang (Fujian Agriculture and Forestry University), who kindly provided the transgenic rice materials, including the OsRhobA-ox line. We also acknowledge professor Ping Wang for his critical comments during the preparation of this report.

## Additional information

### Funding

| Funder | Grant reference number | Author |
|---|---|---|
| Natural Science Foundation of China | 31972979 | Xinyu Liu |
| Mobility program of NSFC and DFG | 31861133017 | Zhengguang Zhang |
| Natural Science Foundation of China | 31671979 | Xiaobo Zheng |
| InnovationTeam Program for NSFC | 2017 | Zhengguang Zhang |

The funders had no role in study design, data collection and interpretation, or the decision to submit the work for publication.

### Author contributions

Xinyu Liu, Conceptualization, Data curation, Software, Formal analysis, Supervision, Funding acquisition, Validation, Investigation, Visualization, Writing - original draft, Project administration, Writing - review and editing; Qikun Zhou, Conceptualization, Data curation, Software, Formal analysis, Funding acquisition, Validation, Visualization, Writing - original draft, Writing - review and editing; Ziqian Guo, Data curation, Software, Formal analysis, Validation, Visualization, Methodology, Writing - original draft; Peng Liu, Data curation, Software, Formal analysis, Validation, Visualization, Methodology; Lingbo Shen, Ning Chai, Data curation, Software, Formal analysis, Validation, Methodology; Bin Qian, Data curation, Software, Validation, Investigation, Methodology; Yongchao Cai, Wenya Wang, Software, Validation, Investigation, Methodology; Ziyi Yin, Conceptualization, Resources, Software, Validation, Writing - review and editing; Haifeng Zhang, Conceptualization, Resources, Formal analysis, Project administration, Writing - review and editing; Xiaobo Zheng, Zhengguang Zhang, Conceptualization, Formal analysis, Supervision, Funding acquisition, Investigation, Project administration, Writing - review and editing

### Author ORCIDs

Zhengguang Zhang (iD) https://orcid.org/0000-0001-8253-4505

### Decision letter and Author response

Decision letter https://doi.org/10.7554/eLife.61605.sa1
Author response https://doi.org/10.7554/eLife.61605.sa2

## Additional files

### Supplementary files
• Supplementary file 1. 574 putative binding proteins of MoAtf1 by CHIP assay. Genomic location and Annotation ID of MoAtf1 binding genes.

• Transparent reporting form

### Data availability
This information can be found in the corresponding figure legends and in the materials and methods section. And the Geo number "GSE144389" for the ChIP assay is available already, we mentioned it in the materials and methods section. All data generated or analysed during this study are included in the manuscript and supporting files. And the information can be found in the corresponding figure legends and in the materials and methods section.

The following dataset was generated:

| Author(s) | Year | Dataset title | Dataset URL | Database and Identifier |
|---|---|---|---|---|
| Liu X, Zhang Z | 2020 | Chip-seq for MoAtf1 | https://www.ncbi.nlm.nih.gov/geo/query/acc.cgi?acc=GSE144389 | NCBI Gene Expression Omnibus, GSE144389 |

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
