## [Decision Letter]

**Acceptance summary:**

The rapid production of reactive oxygen species (ROS) is a widespread defense response in plants. How adapted pathogens overcome this defense response is a fascinating question given that maintaining an appropriate redox balance is critical for pathogen survival throughout infection. Through an extensive set of molecular biology and genetics experiments on the rice blast fungus *Magnaporthe oryzae*, this study uncovers a complete regulatory loop balancing the fungal response to ROS. A cascade of reversible phosphorylation reactions and protein complexes remodeling enable a dynamic response to the redox environment, each step of which contributes to the successful colonization of rice plants by this pathogen.

**Decision letter after peer review:**

[Editors’ note: the authors submitted for reconsideration following the decision after peer review. What follows is the decision letter after the first round of review.]

Thank you for submitting your work entitled "Host-derived ROS activates MoOsm1/MoPtp-dependent phosphorylation of MoAtf1 to orchestra virulence in *Magnaporthe oryzae*" for consideration by *eLife*. Your article has been reviewed by three peer reviewers, one of whom is a member of our Board of Reviewing Editors, and the evaluation has been overseen by a Senior Editor. The reviewers have opted to remain anonymous.

Our decision has been reached after consultation between the reviewers. Based on these discussions and the individual reviews below, we regret to inform you that your work will not be considered further for publication in *eLife*.

We agree that your study is ambitious, but feel that the model you propose is composed of several pieces of evidence that are often incomplete. In some cases, important controls are missing or assumptions are made without justification. While it is admirable to attempt to obtain such a complete view of a signaling node, we suggest you consider focusing on one aspect of your model at a time, in order to put together a more focused and complete study, and submit elsewhere. We are sorry we cannot be more positive at this time.

Reviewer #1:

This manuscript proposes a novel counter-immunity signaling pathway in *Magnaporthe oryzae* during plant infection. This could represent a major advance but substantial revision would be necessary for this to be so. Overall, I felt that evidence was largely correlative rather than causative, conclusions often drawn from what appear to be rather minor differences, and that due-diligence has not been performed for a lot of experiments. This article needs to be revised by an English copy-editor, as the vast number of grammatical errors makes the article difficult to follow. A lot of information is missing from experimental design as described in the text and in figure legends. P-values are mentioned in figure legends but the statistical test is not described.

The Introduction and Discussion are difficult to follow, as many concepts are inadequately described or synthesized.

Are ROS mutants (rbohd or similar) available in the rice cultivars that can be used to validate the conclusion that it is host-derived ROS being detected by the fungus?

Has it been demonstrated that anti-pp38 and anti-p38 are able to recognize MoOSM1? If so please cite. If not, please demonstrate.

Enhanced phosphorylation in Figure 2A is not clear. Perhaps the authors can provide quantification from multiple blots.

Quantification of nuclear:cytoplasmic OsmI1 in Figure 2C-E, and later in Figure 3 is not overly convincing to me. The biological relevance of this sub-cellular movement should be assessed using NES and/or NLS localization/mutant constructs.

Phospho-memetic D/E mutations are not guaranteed to actually mimic phosphorylation. Did the authors also test phospho-ablative A/F mutations? What data support Osm1-Y173D is phosphomimetic? Same is true for Atf1-^S124D^ presented in Figure 5.

The Phostag gel in Figure 4C is not clear, especially when compared to 4F.

In Figure 4E the authors describe “phosphorylation sites” upstream of the NLS: are these just serines, or have they been shown to be phosphorylated? If so, please cite. If not, rephrase.

It is clear that S124 is important for phosphostatus, but there is no evidence that this is due to phosphorylation by Osm1.

IP controls are lacking, and protein expression should be confirmed for the Y2H assays shown in Figure 5. It is possible that the mutant variants are not expressed.

The authors describe “hemibiotrophic fungi” as having a “symbiotic” stage with their plant host. I do not think this is accurate.

The authors demonstrate that ptp1/2 mutants are similar to ptp2 mutants and make the conclusion that PTP1 and PTP2 have differential functions. Are they expressed similarly?

In Figure 7E the legend describes “in vivo” phosphorylation but this described as being performed on protein extracts in the text. Is this really “in vivo”?

What makes a protein/residue hyper-phosphorylated? How can one tell this based on mass spec data without controls? The authors should provide more details about how this MS was performed, and how confident they are in the scores. No Mascot/proteomic confidence scores are provided for these data.

The conclusion that PTP-mediated desphosphorylation of Osm1 leads to nuclear export is not supported by direct evidence and is therefore speculative.

Reviewer #2:

The manuscript by Liu et al. describes molecular mechanisms by which the fungal plant pathogen *Magnaporthe oryzae* copes with reactive oxygen species (ROS) produced during the host immune response. It provides a complete view of one particular mechanism involving phosphorylation/dephosphorylation reactions, protein complex assembly and dissociation, protein relocalization to the nucleus and transcriptional regulation. The manuscript is organized with a straightforward logic, but requires extensive language editing to improve clarity. This is a very ambitious manuscript, featuring a lot of diverse experiments, apparently carefully executed, involving many different molecular players. It therefore has great potential to constitute a significant step forward in our understanding of fungal virulence. However, the relatively complex model the authors attempt to build is composed of pieces of evidence that sometimes remain incomplete. Several steps of the reasoning would require more thorough testing for the model to be fully convincing. An effort is also needed to synthesize these findings and present the conceptual novelty arising from the work. My specific major concerns are as follows:

1) I can see that there is a significant progress being made in understanding a particular ROS-response signaling pathway, the amount of work and new insights appear substantial, but the authors fail to unify their findings and convey the new concepts arising from their experimental work. After reading carefully the manuscript several times, I can tell the authors found new mechanisms for Magnaporthe to sense and respond to host-derived ROS, but I miss a clear statement on what the major novelty/originality is. Can the authors summarize the process they discovered in 1 or 2 simple sentences rather than a list of molecular players? The objectives of the study stated by the authors are:

– "How does *M. oryzae* percept and then overcome ROS stress?" a mechanism is proposed to mitigate the antifungal effect of host-derived ROS is provided, but how ROS are perceived is not directly addressed.

– "how MoOsm1 is linked to MoAp1 or MoAtf1 in stress response is unclear" This is a rather field- or organism-centered question. Some efforts are needed to place the study into a broader perspective an qualify in simple terms the novel concepts and mechanisms discovered.

2) "YFP signal of MoOSM1-cYFP and MoOSM1-nYFP pair was reduced in the nucleus, suggesting that the monomeric form of MoOsm1 is involved in the nuclear localization under the oxidative stress" > Several explanations could fit with this observation, the conclusion is not fully supported. A version of osm1 that remain in the dimer form (maybe a phospho dead mutant of Y173?) should not accumulate in the nucleus.

"Collectively, the results suggested that Y173D but not T171D phosphomimic mutation is required for the interaction"> I would rather conclude that Y173D but not T171D prevents Osm1 oligomer formation

BiFC "the empty vectors used as negative controls" > Better controls would be: Phospho-dead mutants of Osm1 or the Y171D mutant + MoOSM1 wt = positive / MoOsm1Y173D + MoOsm1 wt = negative

Subcellular localisation of MoOsm1Y171D-GFP is not analyzed/commented

3) "MoAtf1 interacts with MoOsm1" and following section> This is lacking a few controls to be fully convincing: what happens if osm is trapped outside the nucleus (nuclear exclusion signal or Y173 phospho dead mutant?) – Since the interaction assay was performed in the absence of ROS stress, what would be its relevance in response to ROS? What would be the function of Osm1 dimer dissociation and relocalization to the nucleus? Is Osm1-Atf1 interaction modified upon ROS stress? Similarly, the interaction between Atf1 and Tup1 has not been tested in the context of response to ROS.

4) "disruption of MoAtf1-MoTup1 complex would reverse the suppression of genes involved in virulence" > as far as I understand, disruption of this complex is achieved through Atf1 phosphorylation (S124D phosphomimic mutant) or deletion of Atf1. However, δ MoATF1 is less virulent than wild type (Figure 4G/H) while there is no Atf1-Tup1 complex and therefore no repression of virulence genes in this strain – Similarly, Tup1 deletion reduced virulence (Chen et al., 2015) – How can we explain these discrepancies? In these experiment, what is the virulence phenotype of Atf1^S124D^ mutants? The reverse controls with phospho-dead mutants of Atf1 would be useful to support the conclusions. In which compartment of the cell (nucleus or cytoplasm) are these interactions happening?

5) Whether the genes under transcriptional control of the Atf1-Tup1 complex actually play a role in virulence is not described. Do they directly contribute to host-derived ROS detoxification?

Reviewer #3:

The article "Host-derived ROS activates MoOsm1/MoPtp-dependent phosphorylation of MoAtf1 to orchestra virulence in *Magnaporthe oryzae*" by Liu et al. propose to demonstrate the role of the MAPK kinase complex MoOsm/MoPtp in the regulation of transcription factors MoAtf1 in the fungus *M. oryzae*.

Overall, authors have provided evidence to show the role of ROS and its effect on one of the MAPK module, but they do not provide evidence to justify the title.

1) Exogenous application of H2O2 is NOT the same as host derived ROS. Authors have not provided evidence to show that host derived ROS can drive the changes in the MAPK complexes in the fungus outlined in this manuscript. Therefore, Authors should use host mutants that cannot make ROS to justify their claim.

2) Authors are aware that fungi have an equivalent ROS producing machinery that can be inhibited by both DPI and catalase (these are not specific to plant NADPH oxidases as claimed by the authors.

3) Authors describe PTI in their Introduction. How and more importantly when is ROS produced during PTI in a suc (LTH) vs res (K23) interaction? They need to define PTI in their pathosystem.

4) Authors need to detail their experimental protocols more thoroughly and edit the figure and figure legends so that they match.

The reviewer questions many of the experimental procedures used:

1) While the Co-IPs are done in hyphae and other time they use protoplasts; many of the localization studies are done in conidia. The authors do not make any distinction between these two. Conidia and hyphae are two distinct developmental stages of fungus as demonstrated by the authors in Figure 10A and authors cannot use interchangeably these two development stages to support their hypothesis

2) With respect to using protoplasts: where the Co-IPs performed from plasmids expressed transiently? No experimental procedure is described with respect to preparation, transformation, and efficiency.

3) What development stage of hyphae was used to extract proteins to perform Co-IPs. How much protein was used in each of the IP experiments?

4) Authors do not provide evidence that validates any of the constructs used in this manuscripts.

5) Monomer/Dimer (Figure 3E). should document this by gel-filtration chromatography and the native gel does NOT adequately demonstrate this.

6) Phosphorylation of MoATF-1 (Figure 4C) – not very convincing (compare 4c to 4F);

Should use ΔMoosm in 4G.

7) EMSA experiments: it is not appropriate it to use a 1 Kb fragment to perform gel shift analysis. An oligomer with the binding site and a mutated binding site should be used.

8) The legend for Figure 9A indicates 100 infecting hyphae were counted. How many of those were used for DAB staining in 9C?

Figures with no explanation or details:

1) Figure 3F: the figure clearly show conidia, but the text refers to it as protoplasts.

2) Figure 1H is referred in text as bulbous structure at 24hrs; not identified in the figure or legend

3) Figure 4H; Figure 5B; middle lane in Figure 7D; what are type 1.…typ4 infection in Figure 9A and 9B?

The reviewer questions some of the statements made in this manuscript:

Results section:

1) MoOsm1 was phosphorylated in response to oxidative stress: "These results suggested that MoOsm1 responds to oxidative stress by inducing a high phosphorylated level and increased accumulation in the nucleus."

This statement should be supported by Western with p-p38 Abs in Figure 2E.

2) MoOsm1 is reduced from a dimer to a monomer under oxidative stress-triggered phosphorylation: "Collectively, the results suggested that Y173D but not T171D phosphomimic mutation is required for the interaction"

Figure 3D clearly shows that wt MoOsm and the mutant MoOsm Y173D do not interact

Appropriate statement would be: phosphorylation of tyrosine 173 inhibits interaction

Figure 3 is a total mess. Legends do not match the Figures and therefore any interpretation from his figure is open to question.

Example: Figure 3 A says: (A) Localization of MoOsm1 and MoOsm1Y173D. The localization of MoOsm1 and MoOsm1Y173D was observed in conidia of the ΔMoosm1 mutants. MoOsm1-GFP and H1-RFP were observed by confocal fluorescence microscopy. Bars = 5μm.

We do not see any localization of MoOsm1 in the nucleus!

In the same legend: A and B is followed by E…..

Then C is refered as a Co-IP assay which clearly it’s not.

3) MoOsm1-triggered MoAtf1 phosphorylation inactivates MoAtf1-MoTup1 Interaction: "We screened MoAtf1-interacting proteins and identified a conserved transcription repressor, MoTup1 (Figure 4—figure supplement 1)."

No explanation or experimental procedures are given with respect to the identification of the interacting proteins. How did they verify that these are genuine interacting proteins?

4) The interaction between MoAtf1 and MoTup1 controls the expression of virulence Factors: "Collectively, the results showed that host-derived ROS accumulation induces the phosphorylation of MoOsm1…."

There is no evidence to support this statement.

5) MoPtps contribute to MoOsm1 dephosphorylation:

"As a hemibiotrophic fungus, *M. oryzae* also maintains a symbiotic relationship with rice without directly killing it."

The fungus has biotrophic phase but no symbiotic relationship.

" As the pathogen utilizes MoOsm1 phosphorylation as a means to control the interaction of MoAtf1-MoTup1 complex in responding to host immunity"

No evidence for this statement.

"We generated respective *ΔMoptp1* and *ΔMoptp2* mutant strains"

"independent *ΔMoptp1ΔMoptp2* double mutants were also constructed"

No evidence is provided for either construction or validation of these mutant strains.

"We further purified the MoOsm1-GFP protein from the ∆Moptp2/MoOSM1-GFP

strain and found that tyrosine 173 was the hyperphosphorylated (Figure 6—figure supplement 1A)."

You can have protein that is hyperphosphorylated, but you cannot have a residue that is hyperphosphorylated…. Constitutively active is more appropriate

6) Identification of MoPtp1 and MoPtp2 as the targets of MoAtf1

"These results indicated that MoAtf1 regulated the expression of MoPTP1 and MoPTP2 by targeting their promoters directly during the growth and development of *M. oryzae*."

No evidence is provided to justify this statement.

[Editors’ note: further revisions were suggested prior to acceptance, as described below.]

Thank you for submitting your article "Host-derived ROS activates a phosphor-regulatory feedback circuitry to govern virulence in *Magnaporthe oryzae*" for consideration by *eLife*. Your article has been reviewed by three peer reviewers, one of whom is a member of our Board of Reviewing Editors, and the evaluation has been overseen by Detlef Weigel as the Senior Editor. The reviewers have opted to remain anonymous.

The reviewers have discussed the reviews with one another and the Reviewing Editor has drafted this decision to help you prepare a revised submission.

Summary:

Plant pathogens face the accumulation of reactive oxygen species (ROS) during the colonization of their host. This study investigates the molecular mechanisms by which the fungal pathogen *Magnaporthe oryzae* responds to ROS during the infection of rice plants. The authors identified a ROS-sensing loop involving protein-protein interactions, subcellular translocation and transcriptional regulation controlled by differential phosphorylation. This thorough investigation proposes a mechanism for restoring fungal cells into a sensitive state after signaling.

Essential revisions:

1) There is no direct evidence that ROS perceived by *M. oryzae* are of plant origin. The authors are asked to provide experimental support for this claim and discuss putative functions of this ROS-sensing mechanism other than counteracting PTI.

2) The manuscript remains difficult to read due to its length and style. Reviewer 3 has made suggestions to streamline the manuscript. The authors should carefully edit the style of their manuscript and make better use of supplementary materials to simplify the main text.

3) Experimental procedures are not always sufficiently explained, justified, and lacking some controls. Please carefully address the reviewer comments related to this concern appended at the bottom of this message.

Reviewer #1:

The manuscript by Liu et al. is a revised version of a study aiming at deciphering the molecular mechanisms by which the fungal plant pathogen *Magnaporthe oryzae* copes with reactive oxygen species produced during the host immune response. I found this new version much improved compared to the previous one. The objectives of the work are much clearer, and the additional experiments improved the quality of the Results section.

There is a number of aspects in which the manuscript should be improved further:

1) Although it improved since the previous version, I would advise the authors need to think of a better title. First, whether ROS are produced by the plant during the interaction cannot be unambiguously established due to some limitations in the experimental system. Therefore the "host-derived" nature of ROS should be downplayed and discussed rather than presented as an established fact. Second, the authors show that ROS dissociate Mosm1 dimers, but it not clear how this is achieved. The use of "activates" may not be the most appropriate in this respect. Third, I feel like "phospho-regulatory circuitry" does not convey fully the originality and breadth of the findings from this manuscript.

2) The Discussion section could be improved by A) commenting on the source of ROS in the interaction. Could they be produced by the fungus instead of the plant? Is there any correlation between the expression of ROS-producing enzymes from Rice and the staining observed in Figure 1? B) commenting on how the current work related to the lifestyle and disease progression (e.g. relationship with hemibiotrophic growth) what is the link with either biotrophic or necrotrophic growth? or rather with virulence in general? C) For discussion: What would be the fitness advantage in recycling Osm1 instead of degrading it when phosphorylated?

3) In spite of real progress, there is still a need to carefully check spelling and grammar throughout the manuscript. (For instance: "To examine MoPtp1/2 in virulence" missing "the role of"; "that of the phosphorylated the MoOsm1-GF" too many "the"; "ROS accumulation around caused by" remove "around"; "pathogenicity assay exhibited that the" > demonstrated that)

Reviewer #2:

The work under revision disentangle the signalling pathway that activates ROS tolerance in a fungal pathogen. This pathway is shown to be critical for virulence and modulate plant immunity during infection. This ambitious work pursues to identify and determine the function of various members of the signalling pathway and the role of each of them in the regulation of ROS tolerance. The work substantially contributes to better understanding how virulence is regulated in plant pathogens. However, I consider that the authors should address the following concerns:

The virulence phenotype of MoTup1 does not fit either with the model. If it is a negative regulator, I would have thought that the knockout would be more virulent. There should be a discussion about this.

Figure 3: I think that there are some problems with the controls: Actin is missing in mutant D in Figure 3B. And also RFP is not detected in mutant A in the nuclear fraction

Finally, Materials and methods section is not complete. For example, I could not find the description on how DAB experiments were performed.

Reviewer #3:

In this manuscript the authors characterized a well conserved signaling pathway in rice blast fungus to show that it responds to changes in ROS, which they try to link to ROS production during immunity. The paper has far too many figures for someone to finish reading it in one go. Also it is a bit difficult to read due to the structuring of sentences.

My main concerns are:

1) I don't think their findings really demonstrate this is a signaling cascade evolved to adapt to counteract the ROS produced during infection. I think the data simply shows Osm1 and its regulators respond to ROS changes in environment. To be able to say it is in response to host produced ROS, they need to have rice NADPH Ox mutants, where Osm1 gains full pathogenesis. I assume this is not possible.

2) Since they are talking about plant infection, the data obtained during appressorium development, which are done on cover slips may not be very relevant. Also, the DPI inhibitor treatments could be problematic. Because the fungus also has NADPH oxidases and they are critical for infection. DPI would probably inhibit fungal enzymes as well.

3) Figure 2: Hydrogen peroxide treatments, the ROS response that the authors see happens 24 hours after inoculation. At that point, the mature appressoria are isolated from the rest of the hyphae. They should repeat these ROS treatment experiments in the right time points to see the response that is relevant to their infection conditions.

4) Do we know if the defects in pathogenesis is due to simple growth defects of the mutants? They should present plate growth assays to show that mutants grow similar to the wild type and only have issues during infection.

5) The text needs to be significantly shortened and focused. There is too much information flowing around, which makes it impossible follow the story. Results section

6) Model: the color code is confusing. On the left, the green squares represent rice cells. On the right, they represent fungal cells.

[Editors' note: further revisions were suggested prior to acceptance, as described below.]

Thank you for resubmitting your work entitled "A host-derived ROS inducible phosphor-regulatory circuitry centering on MoOsm1 governs virulence of *Magnaporthe oryzae*" for further consideration by *eLife*. Your revised article has been evaluated by Detlef Weigel (Senior Editor) , a Reviewing Editor and two reviewers.

The manuscript has been improved but there are some remaining issues that need to be addressed before acceptance, as outlined below:

The additional experiments with OsRbohA lines strengthened the manuscript by providing good support for the described mechanism to function in planta. We believe however that the style of the manuscript still requires attention.

To convey the complex set of results as clearly as possible, we would like to suggest the following revisions to the title and Abstract:

Title: "A self-balancing signaling circuit centered on the Osm1 kinase governs adaptive responses to host-derived reactive oxygen species in *Magnaporthe oryzae*"

Abstract

"The production of reactive oxygen species (ROS) is a ubiquitous defense response in plants. Adapted pathogens evolved mechanisms to counteract the deleterious effects of host-derived ROS and promote infection. How plant pathogens regulate such an elaborate response against ROS burst remains not fully understood. Using the rice blast fungus *Magnaporthe oryzae*, we uncovered a self-balancing circuit controlling response to ROS in planta and virulence. During infection, ROS induces the phosphorylation the high osmolarity glycerol pathway kinase Osm1 and its translocation to the nucleus. There, Osm1 phosphorylates the transcription factor Atf1 and dissociates Atf1-Tup1 complex. This releases Tup1-mediated transcriptional repression on oxidoreduction-pathway genes and activates the transcription of the Ptp1/2 protein phosphatases. In turn, Ptp1/2 dephosphorylate Osm1, restoring the circuit to its initial state. Balanced interactions among Osm1, Atf1, Tup1, and Ptp1/2 proteins provide a means to counter the ROS burst plant immune response. Our findings thereby reveal new insights into how *M. oryzae* utilizes a phosphor-regulatory feedback mechanism to face plant immunity during infection."

Please verify that this provides an accurate account of your work, and please correct and amend it as needed.

---

## [Author Response]

[Editors’ note: the authors resubmitted a revised version of the paper for consideration. What follows is the authors’ response to the first round of review.]

Reviewer #1:This manuscript proposes a novel counter-immunity signaling pathway in Magnaporthe oryzae during plant infection. This could represent a major advance but substantial revision would be necessary for this to be so. Overall, I felt that evidence was largely correlative rather than causative, conclusions often drawn from what appear to be rather minor differences, and that due-diligence has not been performed for a lot of experiments. This article needs to be revised by an English copy-editor, as the vast number of grammatical errors makes the article difficult to follow. A lot of information is missing from experimental design as described in the text and in figure legends. P-values are mentioned in figure legends but the statistical test is not described.The Introduction and Discussion are difficult to follow, as many concepts are inadequately described or synthesized.

We thank the reviewer for the overall positive response to our studies and for the critical but constructive comments. We have addressed all of the concerns raised previously, including data organization and interpretation, experimental design description, and English writing.

Are ROS mutants (rbohd or similar) available in the rice cultivars that can be used to validate the conclusion that it is host-derived ROS being detected by the fungus?

Currently, no ROS defective mutant rice is available to our studies and the common practice in the field has been employing Diphenyleneiodonium (DPI) that inhibits the activity of plant NADPH oxidases, thereby suppressing ROS production (Bolwell et al., 1998; Grant et al., 2000; Zhang et al., 2009). We have routinely treated rice with 0.5 μM DPI to suppress ROS in our studies.

Has it been demonstrated that anti-pp38 and anti-p38 are able to recognize MoOSM1? If so please cite. If not, please demonstrate.

The p38 MAP kinase is the mammalian orthologue of the yeast Hog kinase and the anti-pp38 (CST:9215S) and anti-p38 (CST:9212S) were successfully used for the detection of yeast Hog1. Since MoOsm1 is a Hog1 homolog, we used both antibodies anti-pp38 and anti-p38 to successfully detect MoOsm1 and its phosphorylation. We have since added the description in Materials and methods.

Enhanced phosphorylation in Figure 2A is not clear. Perhaps the authors can provide quantification from multiple blots.

We have repeated the experiments for the quantification (Figure 2D).

Quantification of nuclear:cytoplasmic OsmI1 in Figure 2C-E, and later in Figure 3 is not overly convincing to me. The biological relevance of this sub-cellular movement should be assessed using NES and/or NLS localization/mutant constructs.

Thank you for the suggestion. We have since obtained the MoOsm1-NES strain (which MoOsm1 fused with NES) and provided the localization of both MoOsm1-GFP and MoOsm1-NES strains under H2O2 treatment in Figure 2—figure supplement 1B.

Phospho-memetic D/E mutations are not guaranteed to actually mimic phosphorylation. Did the authors also test phospho-ablative A/F mutations? What data support Osm1-Y173D is phosphomimetic? Same is true for Atf1-^S124D^ presented in Figure 5.

Thank you for the helpful suggestion. In the revised version, MoOsm1 ^Y173A^ strain was obtained. And we evaluated the function of MoOsm1^Y173A^ in the localization and dimer’s formation.

We showed that MoOsm1^Y173D^ was accumulated in the nucleus when compared with MoOsm1 (Figure 3A). Phosphorylation of MoOsm1 was further evaluated using Phos-tag gel electrophoresis. MoOsm1 mobility shift was found in phosphatase treated wild-type cells but not in phosphatase inhibitor-treated cells. A similar band shift was observed in cytoplasmic extracts of the phosphatase treated strain. A decreased mobility shift was found in nuclear MoOsm1-GFP when compared to phosphatase treated strain (Figure 3H). Based on these results, we considered that MoOsm1 ^Y173D^ is phosphomimic that induces the nuclear localization of MoOsm1.

MoAtf1 regulates the transcription of putative target genes (Figure 6 and Supplementary file 1). We also showed that these genes were induced in MoAtf1^S124D^, suggesting that MoAtf1 ^S124D^ is phosphomimic.

The Phostag gel in Figure 4C is not clear, especially when compared to 4F.

We have since improved the presentation by repeating the experiments (Figure 4C).

In Figure 4E the authors describe “phosphorylation sites” upstream of the NLS: are these just serines, or have they been shown to be phosphorylated? If so, please cite. If not, rephrase.

These are serine and threonine sites. We have revised the statements (Figure 4E).

It is clear that S124 is important for phosphostatus, but there is no evidence that this is due to phosphorylation by Osm1.

We showed that S124 is the important phosphorylation site of MoAtf1 (Figure 4F). We then used in vitro phosphorylation analysis to confirm the relationship between MoOsm1 and MoAtf1^S124^ (Figure 4—figure supplement 1). The result validated that MoOsm1 phosphorylates MoAtf1. In addition, when S124 is mutated, phosphorylation of MoAtf1 was reduced significantly, suggesting that S124 is also important for MoOsm1 mediated phosphorylation of MoAtf1.

IP controls are lacking, and protein expression should be confirmed for the Y2H assays shown in Figure 5. It is possible that the mutant variants are not expressed.

IP controls including MoOsm1-NES were provided in the revised version (Figure 4A). For the Y2H assay, we performed PCR to verify the transformants growing on SD-Leu-Trp and SD-Leu-Trp-His-Ade plates. We also used co-IP, BiFC, and GST pull down assays to confirm the interactions.

The authors describe “hemibiotrophic fungi” as having a “symbiotic” stage with their plant host. I do not think this is accurate.

We have since revised the statement.

The authors demonstrate that ptp1/2 mutants are similar to ptp2 mutants and make the conclusion that PTP1 and PTP2 have differential functions. Are they expressed similarly?

The expression levels of *MoPTP1* and *MoPTP2* are shown in Figure 10A. We have improved the description by stating that “Given the possibility that MoPtp1 and MoPtp2 may have redundant functions, independent ∆*Moptp1*∆*Moptp2* double mutants were constructed whose phenotypes are mostly similar to ∆*Moptp2* (Figure 7A and 7B)”.

In Figure 7E the legend describes “in vivo” phosphorylation but this described as being performed on protein extracts in the text. Is this really “in vivo”?

All of the proteins were extracted from *M. oryzae*, with alkaline phosphatase treatment (in vitro) as control, so we deemed that in vivo is more appropriate for the phosphorylation assay.

What makes a protein/residue hyper-phosphorylated? How can one tell this based on mass spec data without controls? The authors should provide more details about how this MS was performed, and how confident they are in the scores. No Mascot/proteomic confidence scores are provided for these data.

We agree and have since revised the statements by providing additional MS information (Figure 5—figure supplement 1).

The conclusion that PTP-mediated desphosphorylation of Osm1 leads to nuclear export is not supported by direct evidence and is therefore speculative.

We have generated the phosphatase domain deletion mutants of MoPtp1 and MoPtp2 and showed that the phosphatase domains were important for MoPtp1 and

MoPtp2 mediated dephosphorylation and also the nuclear exporting of MoOsm1 (Figure 9).

Reviewer #2:The manuscript by Liu et al. describes molecular mechanisms by which the fungal plant pathogen Magnaporthe oryzae copes with reactive oxygen species (ROS) produced during the host immune response. It provides a complete view of one particular mechanism involving phosphorylation/dephosphorylation reactions, protein complex assembly and dissociation, protein relocalization to the nucleus and transcriptional regulation. The manuscript is organized with a straightforward logic, but requires extensive language editing to improve clarity. This is a very ambitious manuscript, featuring a lot of diverse experiments, apparently carefully executed, involving many different molecular players. It therefore has great potential to constitute a significant step forward in our understanding of fungal virulence. However, the relatively complex model the authors attempt to build is composed of pieces of evidence that sometimes remain incomplete. Several steps of the reasoning would require more thorough testing for the model to be fully convincing. An effort is also needed to synthecize these findings and present the conceptual novelty arising from the work. My specific major concerns are as follows:1) I can see that there is a significant progress being made in understanding a particular ROS-response signaling pathway, the amount of work and new insights appear substantial, but the authors fail to unify their findings and convey the new concepts arising from their experimental work. After reading carefully the manuscript several times, I can tell the authors found new mechanisms for Magnaporthe to sense and respond to host-derived ROS, but I miss a clear statement on what the major novelty/originality is. Can the authors summarize the process they discovered in 1 or 2 simple sentences rather than a list of molecular players? The objectives of the study stated by the authors are:– "How does M. oryzae percept and then overcome ROS stress?" a mechanism is proposed to mitigate the antifungal effect of host-derived ROS is provided, but how ROS are perceived is not directly addressed.– "how MoOsm1 is linked to MoAp1 or MoAtf1 in stress response is unclear" This is a rather field- or organism-centered question. Some efforts are needed to place the study into a broader perspective an qualify in simple terms the novel concepts and mechanisms discovered.

We thank the reviewer for the very positive comments regarding our study and for the invaluable comments/suggestions. We have since revised the manuscript based on the reviewers’ comments in various areas to improve the clarity of the manuscript.

2) "YFP signal of MoOSM1-cYFP and MoOSM1-nYFP pair was reduced in the nucleus, suggesting that the monomeric form of MoOsm1 is involved in the nuclear localization under the oxidative stress" > Several explanations could fit with this observation, the conclusion is not fully supported. A version of osm1 that remain in the dimer form (maybe a phospho dead mutant of Y173?) should not accumulate in the nucleus."Collectively, the results suggested that Y173D but not T171D phosphomimic mutation is required for the interaction"> I would rather conclude that Y173D but not T171D prevents Osm1 oligomer formationBiFC "the empty vectors used as negative controls" > Better controls would be: Phospho-dead mutants of Osm1 or the Y171D mutant + MoOSM1 wt = positive / MoOsm1Y173D + MoOsm1 wt = negativeSubcellular localisation of MoOsm1Y171D-GFP is not analyzed/commented

We have since revised the statements to improve data interpretation. We have obtained the MoOsm1^Y173A^ strain and showed that MoOsm1 ^Y173A^ remains in the dimer form and is located in the cytoplasm upon ROS stress (Figure 2A and 2E). We have also employed the recommended BiFC control to evaluate the formation of MoOsm1 dimer under stress (Figure 2G). The localization of MoOsm1Y173D was revealed in Figure 3A.

3) "MoAtf1 interacts with MoOsm1" and following section> This is lacking a few controls to be fully convincing: what happens if osm is trapped outside the nucleus (nuclear exclusion signal or Y173 phospho dead mutant?) – Since the interaction assay was performed in the absence of ROS stress, what would be its relevance in response to ROS? What would be the function of Osm1 dimer dissociation and relocalization to the nucleus? Is Osm1-Atf1 interaction modified upon ROS stress? Similarly, the interaction between Atf1 and Tup1 has not been tested in the context of response to ROS.

We thank the reviewer for the helpful suggestion. By generating a MoOsm1NES mutant strain, we were able to show that the nuclear localization of MoOsm1 is important for its interaction with MoAtf1 (Figure 4A and 4B). We also assessed the interactions under H2O2 treatment that are also present. We reasoned that ROS may not induce all MoOsm1 into phosphor-MoOsm1. Given that deletion of *MoPTP1/2* induces MoOsm1 phosphorylation, similar to being exposed to ROS stress, the interactions between MoOsm1-MoOsm1 and MoAtf1-MoTup1 were detectable in the *Moptp1/Moptp2* mutants. The results showed that the deletion of *MoPTP2* dissociates MoAtf1 from MoTup1 and MoOsm1 dimer formation is blocked in the Δ*Moptp1*Δ*Moptp2* double mutant (Figure 9D and 9E).

4) "disruption of MoAtf1-MoTup1 complex would reverse the suppression of genes involved in virulence" > as far as I understand, disruption of this complex is achieved through Atf1 phosphorylation (S124D phosphomimic mutant) or deletion of Atf1. However, δ MoATF1 is less virulent than wild type (Figure 4G/H) while there is no Atf1-Tup1 complex and therefore no repression of virulence genes in this strain – Similarly, Tup1 deletion reduced virulence (Chen et al., 2015) – How can we explain these discrepancies? In these experiment, what is the virulence phenotype of Atf1^S124D^ mutants? The reverse controls with phospho-dead mutants of Atf1 would be useful to support the conclusions. In which compartment of the cell (nucleus or cytoplasm) are these interactions happening?

We appreciate the insightful comments. We have assessed pathogenicity of MoAtf1^S124D^ and MoAtf1^S124A^ (Figure 4G and 4H) that failed to show any infection differences between wild type (Guy11) and MoAtf1^S124D^ strains on cultivar LTH. The Δ*Moatf1* and Δ*Moatf1*/*MoAtf1^S124A^* mutants exhibited restricted lesions. On cultivar K23, however, wild-type caused few typical lesions (gray spots with brown margins), in contrast to MoAtf1^S124D^ that caused more lesions and larger lesion sizes (Figure 4G and 4H).

Given the function of MoAtf1 and MoTup1 in transcriptional regulation, deletion of MoAtf1 or MoTup1 is expected to result in wide changes in transcription levels. The defects of Δ*Moatf1* and Δ*Motup1* in pathogenicity may be not only solely dependent on MoAtf1-mediated phosphorylation. The conclusion we draw is that MoAtf1 phosphorylation is important for virulence of *M. oryzae*.

5) Whether the genes under transcriptional control of the Atf1-Tup1 complex actually play a role in virulence is not described. Do they directly contribute to host-derived ROS detoxification?

Functions of the putative target genes were predicted and those participate in the oxidoreduction pathway were selected (Figure 6). In MoAtf1 ChIP data, we identified several genes orchestrating host-derived ROS immunity, including MoChia1(MGG_08054) that suppresses chitin-triggered ROS burst by binding chitin.

Reviewer #3:The article "Host-derived ROS activates MoOsm1/MoPtp-dependent phosphorylation of MoAtf1 to orchestra virulence in Magnaporthe oryzae" by Liu et al. propose to demonstrate the role of the MAPK kinase complex MoOsm/MoPtp in the regulation of transcription factors MoAtf1 in the fungus M. oryzae.Overall, authors have provided evidence to show the role of ROS and its effect on one of the MAPK module, but they do not provide evidence to justify the title.1) Exogenous application of H2O2 is NOT the same as host derived ROS. Authors have not provided evidence to show that host derived ROS can drive the changes in the MAPK complexes in the fungus outlined in this manuscript. Therefore, Authors should use host mutants that cannot make ROS to justify their claim.

We thank the reviewer for the insightful comments. Due to limitations in obtaining ROS defective rice cultivars, the research field traditionally employs 0.5 μM diphenyleneiodonium (DPI) to inhibit the activity of plant NADPH oxidases and thereby suppress ROS (Bolwell et al., 1998; Grant et al., 2000; Zhang et al., 2009). We also used DPI to suppress ROS generation in several of our previous studies. Here by treating rice (cultivar K23) with DPI, we found that MoOsm1 phosphorylation was significantly reduced at 24 hpi and 36 hpi (Figure 2C and Results section).

2) Authors are aware that fungi have an equivalent ROS producing machinery that can be inhibited by both DPI and catalase (these are not specific to plant NADPH oxidases as claimed by the authors.

True; both DPI and catalase were applied to rice sheaths prior to being inoculated with *M. oryzae* conidia to minimize fungal exposure to DPI/catalase.

3) Authors describe PTI in their Introduction. How and more importantly when is ROS produced during PTI in a suc (LTH) vs res (K23) interaction? They need to define PTI in their pathosystem.

Previous studies have shown that a moderate resistant cultivar-strain interaction, such as that between K23 and Guy11, produces few and restricted lesions (Liu et al., 2018; Yin et al., 2019b). As NADPH oxidase–mediated ROS production is one of the earliest PTI responses to pathogens, DAB staining is often used to estimate ROS accumulation. Rice cultivar K23 infected with Guy11 yielded reddish-brown precipitates around appressoria and infected hypha at 24, 36, 48, and 60 hpi. Over 40% of infected cells were stained brown at 24 hpi and/or 36 hpi (Figure 1B and 1C).

4) Authors need to detail their experimental protocols more thoroughly and edit the figure and figure legends so that they match.The reviewer questions many of the experimental procedures used:1) While the Co-IPs are done in hyphae and other time they use protoplasts; many of the localization studies are done in conidia. The authors do not make any distinction between these two. Conidia and hyphae are two distinct developmental stages of fungus as demonstrated by the authors in Figure 10 A and authors cannot use interchangeably these two development stages to support their hypothesis

We agreed that conidia and hyphae are the two distinct developmental stages. We used conidia for localization studies because of easy visualization; however, protein extraction from conidia was technically challenging and we thereby used hyphae as the source of protein. Interaction verification by BiFC indicated consistency between conidia and hyphae.

2) With respect to using protoplasts: where the Co-IPs performed from plasmids expressed transiently? No experimental procedure is described with respect to preparation, transformation, and efficiency.

Protoplasts were used for transformation and positive transformants were verified prior to protein extraction. We have provided more details in the Materials and methods section.

3) What development stage of hyphae was used to extract proteins to perform Co-IPs. How much protein was used in each of the IP experiments?

48 hours in liquid CM; this is considered active growing stages for fungal hyphae.

4) Authors do not provide evidence that validates any of the constructs used in this manuscripts.

We have improved the description in the Materials and methods section.

5) Monomer/Dimer (Figure 3E). should document this by gel-filtration chromatography and the native gel does NOT adequately demonstrate this.

We have performed a gel-filtration chromatography assay and the result is shown in Figure 3F.

6) Phosphorylation of MoATF-1 (Figure 4C) – not very convincing (compare 4c to 4F);Should use ΔMoosm in 4G.

We have since repeated the experiment and an improved Figure 4C is presented.

7) EMSA experiments: it is not appropriate it to use a 1 Kb fragment to perform gel shift analysis. An oligomer with the binding site and a mutated binding site should be used.

We have repeated the experiment with the proper marker, as suggested (Figure 10G and 10H).

8) The legend for Figure 9A indicates 100 infecting hyphae were counted. How many of those were used for DAB staining in 9C?

50 infecting hyphae were counted per replicate for the DAB assay. We have added this information to the legend.

Figures with no explanation or details:1) Figure 3F: the figure clearly show conidia, but the text refers to it as protoplasts.

We used conidia for protein localization and since corrected the error we made previously.

2) Figure 1H is referred in text as bulbous structure at 24hrs; not identified in the figure or legend

We have revised the statements in the legend.

3) Figure 4H; Figure 5B; middle lane in Figure 7D; what are type 1.…typ4 infection in Figure 9A and 9B?

Figure 4H, Figure 5B, and Figure 7D have been since revised. We have also added more details in the legends.

The reviewer questions some of the statements made in this manuscript:Results section:1) MoOsm1 was phosphorylated in response to oxidative stress: "These results suggested that MoOsm1 responds to oxidative stress by inducing a high phosphorylated level and increased accumulation in the nucleus."This statement should be supported by Western with p-p38 Abs in Figure 2E.

We appreciate the suggestion. The decreased mobility of MoOsm1-GFP purified from the nucleus was observed when compared to the phosphatase treated strain, indicating a higher level of MoOsm1 phosphorylation in the nucleus (Figure 3H). These results suggest that MoOsm1 could be phosphorylated in the nucleus.

2) MoOsm1 is reduced from a dimer to a monomer under oxidative stress-triggered phosphorylation: "Collectively, the results suggested that Y173D but not T171D phosphomimic mutation is required for the interaction"

We have revised the statements.

Figure 3D clearly shows that wt MoOsm and the mutant MoOsm Y173D do not interactAppropriate statement would be: phosphorylation of tyrosine 173 inhibits interaction

We have revised the statements.

Figure 3 is a total mess. Legends do not match the Figures and therefore any interpretation from his figure is open to question.Example: Figure 3 A says: (A) Localization of MoOsm1 and MoOsm1Y173D. The localization of MoOsm1 and MoOsm1Y173D was observed in conidia of the ΔMoosm1 mutants. MoOsm1-GFP and H1-RFP were observed by confocal fluorescence microscopy. Bars = 5μm.We do not see any localization of MoOsm1 in the nucleus!In the same legend: A and B is followed by E…..Then C is refered as a Co-IP assay which clearly it’s not.

We have revised the statements. We have provided Western blot analysis and the fluorescence intensity assay to evaluate the localization of MoOsm1 under H2O2 treatment (Figure 3).

3) MoOsm1-triggered MoAtf1 phosphorylation inactivates MoAtf1-MoTup1Interaction: "We screened MoAtf1-interacting proteins and identified aconserved transcription repressor, MoTup1 (Figure 4—figure supplement 1)."No explanation or experimental procedures are given with respect to the identification of the interacting proteins. How did they verify that these are genuine interacting proteins?

All of the proteins were identified by utilizing the affinity purification approach. Following affinity purification, proteins bound to anti-GFP beads were eluted and analyzed by mass spectrometry (MS). We have provided more information on MoAtf1-interaction proteins in Figure 5—figure supplement 1.

4) The interaction between MoAtf1 and MoTup1 controls the expression of virulenceFactors: "Collectively, the results showed that host-derived ROS accumulationinduces the phosphorylation of MoOsm1…."There is no evidence to support this statement.

We agree and have assessed MoOsm1 phosphorylation during infection under DPI treatment. The results showed that, when K23 was treated with 0.5 μM DPI, MoOsm1 phosphorylation was significantly attenuated at 24 hpi and 36 hpi (Figure 2C and Results section).

5) MoPtps contribute to MoOsm1 dephosphorylation:

We have performed a series of experiments to support this conclusion. We generated phosphatase activity inactivation mutants of MoPtp1 and MoPtp2 (∆*Moptp1*/*MoPTP1*^∆*ptpc*^ and ∆*Moptp2*/*MoPTP2*^∆*ptpc*^mutant) and evaluated their MoOsm1 phosphorylation pattern. The results showed that in both ∆*Moptp1*/*MoPTP1*^∆*ptpc*^ and ∆*Moptp2*/*MoPTP2*^∆*ptpc*^mutants, the band of MoOsm1-GFP migrated as the phosphorylated MoOsm1-GFP protein treated with an inhibitor (also in ∆*Moptp1/2* mutants (Figure 9B)), indicating that the phosphatase activity is critical for the dephosphorylation of MoOsm1. We also detected the phosphorylation levels of MoOsm1 using the antiphospho-p38 antibody, and observed an increased MoOsm1 phosphorylation in both Δ*Moptp1* and Δ*Moptp2* mutants (Figure 7—figure supplement 2). In addition, the phosphorylation of MoOsm1 remains at high levels even after H2O2 treatment at 60 minutes in the Δ*Moptp2* mutants, while the wild type decreased to a low level at 10 minutes (Figure 7—figure supplement 2). We then observed the localization of MoOsm1 in the Δ*Moptp1*, Δ*Moptp2,* ∆*Moptp1*/*MoPTP1*^∆*ptpc*^, and ∆*Moptp2*/*MoPTP2*^∆*ptpc*^mutants. MoOsm1 was present in both the cytosol and the nucleus evenly, similar to that in wild type. When treated with H2O2, MoOsm1 shows an enhanced nuclear translocation pattern. At 30 minutes following H2O2 treatment, MoOsm1 shows a nucleus to cytoplasm shifting in the wild type strain but not in the Δ*Moptp*2 and ∆*Moptp2*/*MoPTP2*^∆*ptpc*^ mutants (Figure 9C). These results further supported that MoPtp1/2-mediated dephosphorylation of MoOsm1 controls its nuclear-cytoplasm translocation.

"As a hemibiotrophic fungus, M. oryzae also maintains a symbiotic relationshipwith rice without directly killing it."The fungus has biotrophic phase but no symbiotic relationship.

We have corrected the error.

"As the pathogen utilizes MoOsm1 phosphorylation as a means to control the interaction of MoAtf1-MoTup1 complex in responding to host immunity"No evidence for this statement.

We have made the proper correction.

"We generated respective ΔMoptp1 and ΔMoptp2 mutant strains""independent ΔMoptp1ΔMoptp2 double mutants were also constructed"No evidence is provided for either construction or validation of these mutant strains.

Southern blot analysis was performed for both Δ*Moptp1* and Δ*Moptp2* mutant strains (Figure 6—figure supplement 1).

"We further purified the MoOsm1-GFP protein from the ∆Moptp2/MoOSM1-GFPstrain and found that tyrosine 173 was the hyperphosphorylated (Figure 6—figure supplement 1A)."You can have protein that is hyperphosphorylated, but you cannot have a residue that is hyperphosphorylated…. Constitutively active is more appropriate

We have made the proper correction.

6) Identification of MoPtp1 and MoPtp2 as the targets of MoAtf1"These results indicated that MoAtf1 regulated the expression of MoPTP1 and MoPTP2 by targeting their promoters directly during the growth and development of M. oryzae."No evidence is provided to justify this statement.

We have revised the statements to “These results indicated that MoAtf1 regulates the expression of *MoPTP1* and *MoPTP2* by directly targeting their promoters”.

[Editors’ note: what follows is the authors’ response to the second round of review.]

Essential revisions:1) There is no direct evidence that ROS perceived by M. oryzae are of plant origin. The authors are asked to provide experimental support for this claim and discuss putative functions of this ROS-sensing mechanism other than counteracting PTI.

We appreciate the critical comments. Previous studies demonstrate that RbohA functions as the NADPH oxidase critical for ROS generation in rice, and OsRbohA-overexpressing transgenic plants exhibit higher ROS production (Wang et al., 2016). We have provided the evidence that MoOsm1 phosphorylation levels were induced in the OsRbohA mutant (Figure 2D). And we have also shown that the treatment of DPI, which inhibits ROS generation in rice, resulted in a significant decrease in MoOsm1 phosphorylation (Figure 2C). These results revealed that MoOsm1 indeed responds to ROS stress during infection by *M. oryzae*. We have improved relevant statements in the revised version.

2) The manuscript remains difficult to read due to its length and style. Reviewer 3 has made suggestions to streamline the manuscript. The authors should carefully edit the style of their manuscript and make better use of supplementary materials to simplify the main text.

We appreciate the helpful suggestion and have improved manuscript presentation through English language editing.

3) Experimental procedures are not always sufficiently explained, justified, and lacking some controls. Please carefully address the reviewer comments related to this concern appended at the bottom of this message.

We have improved the presentation by adding additional descriptions and literature citation.

Reviewer #1:The manuscript by Liu et al. is a revised version of a study aiming at deciphering the molecular mechanisms by which the fungal plant pathogen Magnaporthe oryzae copes with reactive oxygen species produced during the host immune response. I found this new version much improved compared to the previous one. The objectives of the work are much clearer, and the additional experiments improved the quality of the Results section.There is a number of aspects in which the manuscript should be improved further:1) Although it improved since the previous version, I would advise the authors need to think of a better title. First, whether ROS are produced by the plant during the interaction cannot be unambiguously established due to some limitations in the experimental system. Therefore the "host-derived" nature of ROS should be downplayed and discussed rather than presented as an established fact. Second, the authors show that ROS dissociate Mosm1 dimers, but it not clear how this is achieved. The use of "activates" may not be the most appropriate in this respect. Third, I feel like "phospho-regulatory circuitry" does not convey fully the originality and breadth of the findings from this manuscript.

Thank you for the helpful suggestion. Following new evidence of OsRboh overexpression inducing ROS generation and MoOsm1 phosphorylation, we thought that a “inducible” may better depict MoOsm1-MoAtf1-MoPtps in the phosphor-regulatory process, and thereby propose “A host-derived ROS inducible phosphor-regulatory circuitry centering on MoOsm1 governs virulence of *Magnaporthe oryzae*” as an updated title.

2) The Discussion section could be improved by A) commenting on the source of ROS in the interaction. Could they be produced by the fungus instead of the plant? Is there any correlation between the expression of ROS-producing enzymes from Rice and the staining observed in Figure 1? B) commenting on how the current work related to the lifestyle and disease progression (e.g. relationship with hemibiotrophic growth) what is the link with either biotrophic or necrotrophic growth? or rather with virulence in general? C) For discussion: What would be the fitness advantage in recycling Osm1 instead of degrading it when phosphorylated?

We appreciate the insightful comments. In the revised Discussion section, we discussed the following aspects:

A) For the pathogen, ROS seems like a double-edged sword. During appressorial penetration, *M. oryzae* accumulates high levels of endogenous ROS to strengthen the appressorium cell wall. Here, ROS accumulation is regulated by two fungal NADPH oxidases, Nox1 and Nox2, which themselves are important for appressorium-mediated cuticle penetration. However, ROS is also one of the earliest responses by plants to microbial colonization. This oxidative burst function a potent defense mechanism that limits fungal biotrophic growth (Torres, 2010). Rboh, which encodes NADPH oxidase, is the key host protein that generates ROS upon the pathogen challenge (Wong et al., 2007). Both OsrbohA and OsrbohB are important for this process (Wong et al., 2007). We have updated the relevant description.

B) During the interaction between *M. oryzae* and the host, the host induces ROS burst during the biotrophic growth stage of the fungus. Once sensing this stress, the pathogen activates MoOsm1-mediated pathways that in turn activates the transcription factor MoAtf1. The phosphorylation of MoAtf1 causes the disintegration of the MoAtf1-MoTup1 complex to induce the expression of oxidation regulation pathway genes to further enhance virulence of *M. oryzae*. Phosphorylated MoAtf1 also initiates the transcription of the *MoPTP1/2* genes to further dephosphorylate MoOsm1. Once the ROS stress was circumvented, such as that at the necrotrophic growth stage, MoPtps mediated dephosphorylation of MoOsm1 shut off phosphor-regulatory circuitry to control the virulence.

C) Given the evidence indicating that the nuclear localization of MoOsm1 was more stable in the Δ*Moptps* mutant than Guy11 under ROS stress, it is plausibly that MoPtps function in the recycling of MoOsm1 to the cytoplasm. MoOsm1 recycling may have two benefits: 1) shutdown of phosphorylated MoOsm1 mediates signaling pathways that switch off virulence attack. 2) recycled MoOsm1 in the cytoplasm might respond quickly to external stress as there is no need for protein resynthesing. We have updated the relevant statements.

3) In spite of real progress, there is still a need to carefully check spelling and grammar throughout the manuscript. (For instance: "To examine MoPtp1/2 in virulence" missing "the role of"; "that of the phosphorylated the MoOsm1-GF" too many "the"; "ROS accumulation around caused by" remove "around"; "pathogenicity assay exhibited that the" > demonstrated that)

We have done so in the revised manuscript.

Reviewer #2:The work under revision disentangle the signalling pathway that activates ROS tolerance in a fungal pathogen. This pathway is shown to be critical for virulence and modulate plant immunity during infection. This ambitious work pursues to identify and determine the function of various members of the signalling pathway and the role of each of them in the regulation of ROS tolerance. The work substantially contributes to better understanding how virulence is regulated in plant pathogens. However, I consider that the authors should address the following concerns:The virulence phenotype of MoTup1 does not fit either with the model. If it is a negative regulator, I would have thought that the knockout would be more virulent. There should be a discussion about this.

Studies revealed that Tup1 is a critical transcription repressor that does not bind directly to DNA but is brought to the promoters via interactions with sequence-specific regulatory proteins to regulate the expression of genes. Here, the disrupted interaction between MoAtf1 and MoTup1 induces the expression of various genes. However, MoTup1 may regulate various other transcription factors in addition to MoAtf1, so it is plausible that the deletion of *MoTUP1* causes defect in virulence expression.

Figure 3: I think that there are some problems with the controls: Actin is missing in mutant D in Figure 3B. And also RFP is not detected in mutant A in the nuclear fractionFinally, Materials and methods section is not complete. For example, I could not find the description on how DAB experiments were performed.

We have provided additional information in the Materials and methods, including the DAB assay.

Reviewer #3:In this manuscript the authors characterized a well conserved signaling pathway in rice blast fungus to show that it responds to changes in ROS, which they try to link to ROS production during immunity. The paper has far too many figures for someone to finish reading it in one go. Also it is a bit difficult to read due to the structuring of sentences.My main concerns are:1) I don't think their findings really demonstrate this is a signaling cascade evolved to adapt to counteract the ROS produced during infection. I think the data simply shows Osm1 and its regulators respond to ROS changes in environment. To be able to say it is in response to host produced ROS, they need to have rice NADPH Ox mutants, where Osm1 gains full pathogenesis. I assume this is not possible.

Thank you for the helpful suggestion. OsRbohA is an important NADPH oxidase critical for ROS generation in rice. Here, Prof. Kunming Cheng kindly provided us with the OsRbohA over-expression strain that allowed us to detect MoOsm1 phosphorylation during the infection (Figure 2D and subsection “MoOsm1 phosphorylation in response to oxidative stress”).

2) Since they are talking about plant infection, the data obtained during appressorium development, which are done on cover slips may not be very relevant. Also, the DPI inhibitor treatments could be problematic. Because the fungus also has NADPH oxidases and they are critical for infection. DPI would probably inhibit fungal enzymes as well.

Thank you for the help comments. *M. oryzae* produces the appressorium during infection and the evidence we provided were performed on rice infected with *M. oryzae* for 8 h (Figure 2). To further confirm the result on appressorium development, we provided the results at 20 hpi post infection on the revised version. We used the DPI inhibitor to treat rice sheath before infection with *M. oryzae,* and we agree that the fungus also produces NADPH oxidases critical for infection too. However, we showed that the wild type infection hyphae could expend into adjacent rice cells when treated with 0.5 μm DPI, indicating that DPI causes no defects on the infection ability of *M. oryzae*.

3) Figure 2: Hydrogen peroxide treatments, the ROS response that the authors see happens 24 hours after inoculation. At that point, the mature appressoria are isolated from the rest of the hyphae. They should repeat these ROS treatment experiments in the right time points to see the response that is relevant to their infection conditions.

We agreed that hydrogen peroxide treatment was not similar to that of infection. At 24 hpi, *M. oryzae* penetrated the epidermis and infected the host cells (Figure 1). As it is difficult to treat with rice sheath cell with H2O2 in vitro, we have provided the evidence that the overexpression line of OsRboha shows significant induced MoOsm1 phosphorylation levels than that of WT, indicating that ROS treatment induces MoOsm1 phosphorylation. In addition, we tried to treat appressoria with 5mM H2O2, however, it destroys appressoria. We thereby provide the results at 20 hpi post infection to confirm the phosphorylation levels of MoOsm1 at the appressorial stages.

4) Do we know if the defects in pathogenesis is due to simple growth defects of the mutants? They should present plate growth assays to show that mutants grow similar to the wild type and only have issues during infection.

We have provided the evidence through revision (Figure 6—figure supplement 2).

5) The text needs to be significantly shortened and focused. There is too much information flowing around, which makes it impossible follow the story.

We appreciate the helpful suggestions and have updated relevant descriptions, including the title.

6) Model: the color code is confusing. On the left, the green squares represent rice cells. On the right, they represent fungal cells.

Revised.

[Editors' note: further revisions were suggested prior to acceptance, as described below.]

The manuscript has been improved but there are some remaining issues that need to be addressed before acceptance, as outlined below:The additional experiments with OsRbohA lines strengthened the manuscript by providing good support for the described mechanism to function in planta. We believe however that the style of the manuscript still requires attention.To convey the complex set of results as clearly as possible, we would like to suggest the following revisions to the title and Abstract:Title: "A self-balancing signaling circuit centered on the Osm1 kinase governs adaptive responses to host-derived reactive oxygen species in Magnaporthe oryzae"Abstract"The production of reactive oxygen species (ROS) is a ubiquitous defense response in plants. Adapted pathogens evolved mechanisms to counteract the deleterious effects of host-derived ROS and promote infection. How plant pathogens regulate such an elaborate response against ROS burst remains not fully understood. Using the rice blast fungus Magnaporthe oryzae, we uncovered a self-balancing circuit controlling response to ROS in planta and virulence. During infection, ROS induces the phosphorylation the high osmolarity glycerol pathway kinase Osm1 and its translocation to the nucleus. There, Osm1 phosphorylates the transcription factor Atf1 and dissociates Atf1-Tup1 complex. This releases Tup1-mediated transcriptional repression on oxidoreduction pathway genes and activates the transcription of the Ptp1/2 protein phosphatases. In turn, Ptp1/2 dephosphorylate Osm1, restoring the circuit to its initial state. Balanced interactions among Osm1, Atf1, Tup1, and Ptp1/2 proteins provide a means to counter the ROS burst plant immune response. Our findings thereby reveal new insights into how M. oryzae utilizes a phosphor-regulatory feedback mechanism to face plant immunity during infection."Please verify that this provides an accurate account of your work, and please correct and amend it as needed.

Thank you for the critical comment on the title and Abstract. We agreed that the new title and Abstract convey the results much more clearly. In view of the word limitation of *eLife*, we made some modification to the title into “A self-balancing circuit centered on MoOsm1 kinase governs adaptive responses to host-derived ROS in *Magnaporthe oryzae*”.

In the Abstract, the new Abstract provides the accurate account of our work. As we used the *M. oryzae* as a model organism, we thought MoOsm1(MoAtf1, MoTup1 and MoPtp) should instead of Osm1(Atf1, Tup1 and Ptp) in the Abstract. We have revised the text.